# A temporal sequence of thalamic activity unfolds at transitions in behavioral arousal state

Beverly Setzer[1,2], Nina E. Fultz [2,3], Daniel E. P. Gomez[2,3,4], Stephanie D. Williams[2], Giorgio Bonmassar[3,4], Jonathan R. Polimeni[3,4,5] & Laura D. Lewis [2,3] ✉

Awakening from sleep reflects a profound transformation in neural activity and behavior. The thalamus is a key controller of arousal state, but whether its diverse nuclei exhibit coordinated or distinct activity at transitions in behavioral arousal state is unknown. Using fast fMRI at ultra-high field (7 Tesla), we measured sub-second activity across thalamocortical networks and within nine thalamic nuclei to delineate these dynamics during spontaneous transitions in behavioral arousal state. We discovered a stereotyped sequence of activity across thalamic nuclei and cingulate cortex that preceded behavioral arousal after a period of inactivity, followed by widespread deactivation. These thalamic dynamics were linked to whether participants subsequently fell back into unresponsiveness, with unified thalamic activation reflecting maintenance of behavior. These results provide an outline of the complex interactions across thalamocortical circuits that orchestrate behavioral arousal state transitions, and additionally, demonstrate that fast fMRI can resolve sub-second subcortical dynamics in the human brain.

Striking behavioral changes accompany arousal from sleep, as we transition from unresponsiveness to engaging with our sensory environment. These behavioral changes reflect a fundamental shift in neural dynamics throughout the cortex and subcortex. Extensive studies of arousal state transitions have identified their characteristic electroencephalography (EEG) signatures[1–4], as arousal is marked by changes in brain rhythms[2,5–10] and functional connectivity[11–14] throughout thalamocortical networks. Although the differences in neural dynamics between stable states of sleep and wakefulness have been well characterized, how thalamocortical networks implement transitions between behavioral states is not well understood. Furthermore, while arousals are often characterized by the EEG, behavior is not always linked to a specific EEG signature[15–17], which leaves open the question of which neural dynamics mediate behavioral state transitions. In particular, the specific sequence of neural activity that

unfolds at the moment of behavioral state transitions has not yet been mapped throughout large-scale thalamocortical networks.

The thalamus is a key controller of arousal state transitions. Animal studies have shown that brainstem, hypothalamic, and basal forebrain nuclei can causally control sleep-wake transitions[18–25], and have identified the thalamus as a convergent region through which they regulate cortical state. The thalamus' profuse cortical interconnections aptly position it to alter whole-brain dynamics, as it acts as a central hub for information relay and regulation[24–27] and coordinates cortical rhythms[25,28–30]. Furthermore, direct stimulation of individual thalamic nuclei has been shown to drive arousal state transitions[24,30–42]. Human imaging studies have further confirmed that spontaneous thalamic activity shifts across arousal state: functional magnetic resonance imaging (fMRI) studies have shown altered thalamocortical and intrathalamic connectivity during sleep[43–46], and distinct

[1]Graduate Program for Neuroscience, Boston University, Boston, MA 02215, USA. [2]Department of Biomedical Engineering, Boston University, Boston, MA 02215, USA. [3]Athinoula A. Martinos Center for Biomedical Imaging, Massachusetts General Hospital, Charlestown, MA 02129, USA. [4]Department of Radiology, Harvard Medical School, Boston, MA 02115, USA. [5]Division of Health Sciences and Technology, Massachusetts Institute of Technology, Cambridge, MA, USA. ✉e-mail: ldlewis@bu.edu

spatiotemporal dynamics linked to arousal state fluctuations[47–49]. Intriguingly, the thalamus has also been shown to activate during arousal state transitions in humans[50,51].

The thalamus is made up of many diverse nuclei with unique structural and functional properties[52–54], and whether their individual activity is coordinated or distinct at state transitions has not yet been established. Invasive studies have provided important insights into the causal role of several nuclei of the thalamus, showing that individual nuclei can modulate arousal. However, the joint dynamics across multiple nuclei are not well understood, as addressing this question requires recording from many nuclei simultaneously. Traditional non-invasive methods such as conventional fMRI can provide whole-brain simultaneous imaging, but previously lacked the temporal resolution necessary to capture fast dynamics. A key open question is whether behavioral state transitions are due to synchronous changes throughout thalamocortical networks, or a specific cascade of neural events.

In this work, we use high temporal resolution (fast) fMRI to delineate the thalamocortical dynamics underlying behavioral arousals. Advances in pulse sequence technologies now enable acquiring whole-brain fMRI data at fast rates[55–57] (e.g., repetition time (TR) under 400 ms), and investigations of the hemodynamic response have demonstrated that fast fMRI data can resolve neural dynamics on timescales of hundreds of milliseconds in cortex and thalamus[58–65]. We exploit these fast fMRI techniques to track rapid thalamic dynamics at the moment when the behavioral arousal state changes. We aim to identify the neural dynamics that underlie the recovery of behavior after prolonged unresponsiveness, and refer to this transition in behavioral state as "arousal". First, we use fast fMRI at 3 T to investigate the moment of arousal and find thalamic activation prior to arousal. Then, to delineate activity within distinct thalamic nuclei, we collect a second dataset using fast fMRI at 7 T to image nine individual thalamic nuclei with sub-second resolution. We identify a temporal sequence of activity that unfolds across thalamic nuclei just prior to the moment of behavioral arousal, and further determine that this sequence precedes a brain-wide reorganization of thalamocortical dynamics that reflects subsequent behavioral state. These results precisely identify a cascade of neural events that unfold as the brain switches between behavioral arousal states.

## Results

### Identifying moments of behavioral arousal state transitions

We aimed to identify the thalamocortical dynamics underlying behavioral arousal state transitions by imaging participants spontaneously transitioning between sleep and wakefulness during nighttime fMRI scans. Arousal from sleep occurs spontaneously throughout the night[66], and we observed occasional spontaneous awakenings in participants sleeping in the scanner. To track behavioral arousal state without a stimulus that would influence arousal directly, all subjects performed a self-paced behavioral task in which they pressed a button with every breath[17,67]. Arousal was defined as the first button-press after at least 20 s of unresponsiveness. Because movement sometimes accompanies arousal from sleep, arousals with high motion (>0.3 mm) were excluded to minimize fMRI artifacts.

With this approach, we identified 66 arousals from six subjects (mean = 11, std = 11.78, min = 1, max = 29, Supplementary Table 1) with simultaneous EEG and fast fMRI at 3 T in Experiment 1. Traditional sleep-scoring from EEG recordings revealed that most of these arousals occurred from NREM sleep: 15 s before the arousal event, 20% were awake, 3% were in N1 sleep, 51% were in N2 sleep, and 26% were in N3 sleep (Supplementary Fig. 1a). The rigid 30-s intervals used in sleep-scoring do not always reflect the transient dynamics of arousal state transitions, and since some of the segments classified as awake likely corresponded to N1 sleep, the proportion of behavioral arousals representing transitions out of sleep may have been higher than the

observed 80%. However, this result also demonstrated that behavioral arousal state transitions could occur at low rates during EEG-defined wakefulness.

We calculated the alpha (8–13 Hz) power in the occipital electrode to evaluate electrophysiological dynamics coupled to these behavioral arousals, because occipital alpha defines wakefulness in polysomnography[2]. We found that behavioral arousal was accompanied by a rise in occipital alpha power (Fig. 1; $p < 0.05$; paired $t$-test), consistent with its link to awakening from sleep. Though behavioral and electrophysiological measures provide different definitions of arousal[16,17], these results confirmed that our definition of behavioral arousal captured a state transition linked to—but not identical to—awakening from EEG-defined sleep.

### Thalamus activates before transitions in behavioral arousal state

Our fast fMRI acquisition (TR = 367 ms at 3 T in Experiment 1) enabled us to delineate temporally precise signals within the thalamus and cortex during arousal. We first investigated the dynamics of the whole thalamus and whole cortex by extracting the mean arousal-locked fMRI time series of each region. Consistent with prior studies, we found an overall increase in thalamic activity and a decrease in cortical activity (Fig. 2a), locked to arousal[50,51,68]. However, examining the time courses in each region revealed distinct temporal dynamics: after a slight rise in both regions, the thalamus exhibited a striking increase in activity before arousal. After arousal, cortical signals sharply declined,

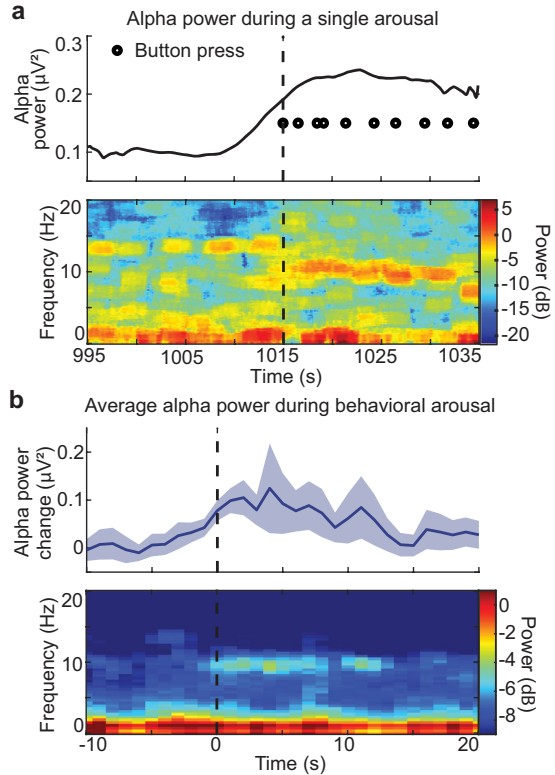

**Fig. 1 | Behavioral arousals are linked to a transition in cortical EEG. a** An example behavioral arousal (dashed line) is accompanied by an increase in occipital alpha power (8–13 Hz). Top: the alpha power increase coincides with the return of behavior. Power is temporally smoothed in 2 s sliding windows. Bottom: the spectrogram demonstrates a return of occipital alpha directly following behavioral arousal. **b** Top: the mean alpha power significantly increases at behavioral arousal, consistent with a transition to the awake state (6 subjects, 66 arousals). Power is temporally smoothed in 2 s sliding windows. Shading is a standard error. Bottom: average spectrogram shows a specific increase in alpha power during behavioral arousal. Source data are provided in "Fig. 1 Source Data" file.

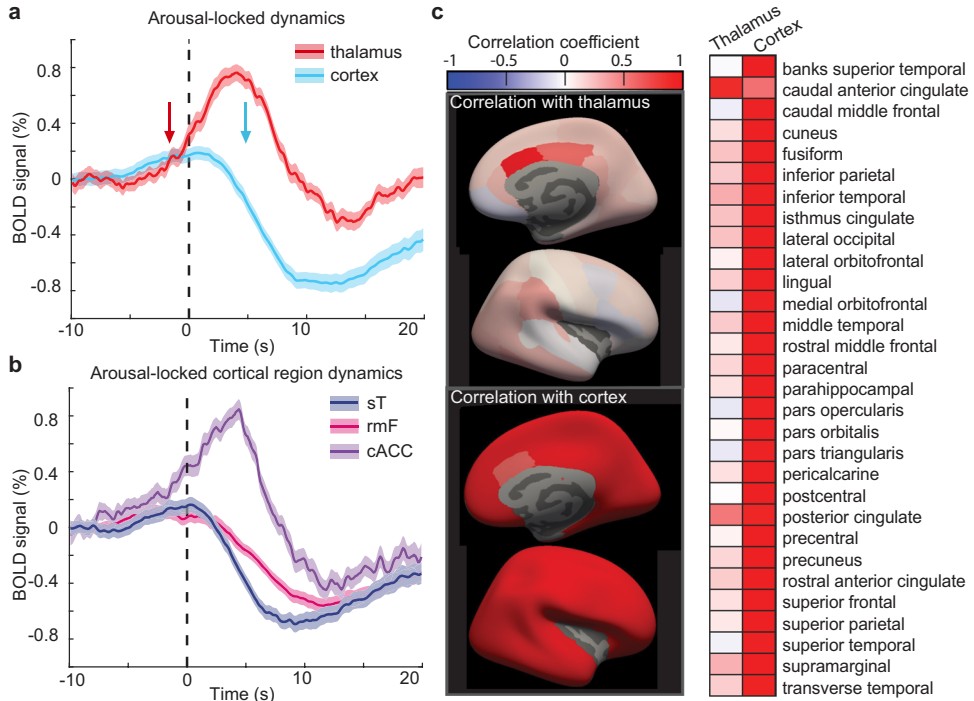

**Fig. 2 | Thalamus activates seconds before behavioral arousal, while most of the cortex deactivates after arousal. a** Arousal-locked blood oxygenation-level-dependent (BOLD) signals in the whole cortex and thalamus. Thalamus (red) begins to activate (20% latency denoted by red arrow) seconds before behavioral arousal (vertical dashed line), while the cortex (blue) deactivates afterward (blue arrow). Data were presented as mean values, and shading represents standard error. **b** Most of the cortex deactivates during arousal, but a subset of regions, including the caudal anterior cingulate cortex (cACC, purple), activate before behavioral arousal,

similarly to the thalamus. The rostral middle frontal (rmF, pink) and superior temporal (sT, dark blue) are shown as representative regions of frontal and temporal lobes. Results from all cortical ROIs can be seen in Supplementary Fig. 2. Shading represents standard error. **c** Correlation between each individual cortical region and the whole cortex or thalamus. While cortical activity is largely distinct from the thalamic rise, a subset of cortical regions is correlated with the thalamus, notably the caudal anterior cingulate and the posterior cingulate. Source data are provided in the "Fig. 2 Source Data" file.

and thalamic signals returned to baseline. Notably, the latency and waveform of the arousal-locked fMRI signatures differed across the thalamus and cortex. The thalamus activated before behavioral arousal, with a latency (defined as the time at 20% of maximum absolute signal) of −1.65 s before arousal (95% confidence interval, CI: (−3.62 s, 0 s)), while the cortex deactivated afterward with a latency of 4.77 s after arousal (95% CI: (3.76 s, 5.69 s)). These results demonstrated that the increase in thalamic activity began seconds earlier than the return of behavior, and diverged from the pattern seen in the global cortical signal.

Because Experiment 1 recorded simultaneous EEG and fMRI, we were able to identify the sleep stage prior to behavioral arousal. While most arousals were clearly from deeper stages of sleep (N2 and N3), our data did include some arousals from light N1 sleep and wakefulness (Supplementary Fig. 1a). Therefore, we investigated whether this same thalamocortical pattern was present when analyzing only arousals from the deeper stages of sleep, yielding 50 arousals from five subjects. We found that the arousal-locked signatures in the thalamus and cortex were broadly conserved (Supplementary Fig. 1b). The thalamus activated −1.65 s before arousal (20% latency; 95% CI: (−3.58 s, 0 s)), and the cortex deactivated 4.77 s after arousal (20% latency = 95% CI: (3.67 s, 5.50 s)). Thus, we concluded that the moment of arousal was preceded by a large increase in thalamic activity and a subsequent decrease in cortical activity.

To investigate whether these arousal-locked patterns were present throughout the cortex or within specific areas, we used the Desikan–Killiany atlas[69] to extract 30 cortical regions of interest (ROIs). The vast majority of the cortex did not show a significant increase in arousal (26/30, two-sided t-test Bonferroni corrected); however, there were a few notable exceptions (Fig. 2b and Fig. S2), including the caudal

anterior cingulate cortex (cACC) and posterior cingulate cortex (PCC). The cACC and PCC activity increased before arousal (20% latency = −3.30 s, 95% CI: (−6.19 s, −0.78 s) in cACC, and 20% latency = −5.23 s, CI: (−6.70 s, −2.57 s) in PCC) and showed higher correlations with the thalamic arousal-locked fMRI signal than other cortical regions (Fig. 2c). The cingulate cortex thus had unique arousal-locked activity within the cortex, resembling the thalamus in its profile of an early, large activation that preceded the moment of behavioral arousal.

## Behavioral arousal-locked dynamics across individual thalamic nuclei and cortical regions

Our results confirmed a striking increase in thalamic activity at the moment of behavioral arousal and further identified the temporal dynamics of this activity, showing that thalamic signals precede behavioral state changes and cortical deactivation by several seconds. However, the thalamus is a heterogeneous structure made up of many functionally distinct nuclei, and their respective roles in arousal are not well understood. The temporal signal-to-noise ratio of individual thalamic nuclei is low at 3 T. Therefore, we repeated our study using 7 T fMRI to enhance subcortical signals and enable imaging of individual thalamic nuclei in Experiment 2. Using the same nighttime scanning procedure and behavioral task, we recorded 97 behavioral arousals from 13 subjects (mean = 7.46 per subject, std = 5.55, min = 1, max = 21, Supplementary Table 2). We extracted thalamic nuclei by segmenting them anatomically in individual subject space, allowing us to identify specific nuclei within each individual[68]. To minimize signal blurring across nuclei, we used a conservative definition and analyzed only the thalamic nuclei which had a 90% probability or higher of filling at least one functional voxel in all subjects[70], yielding nine nuclei of interest (Fig. 3a).

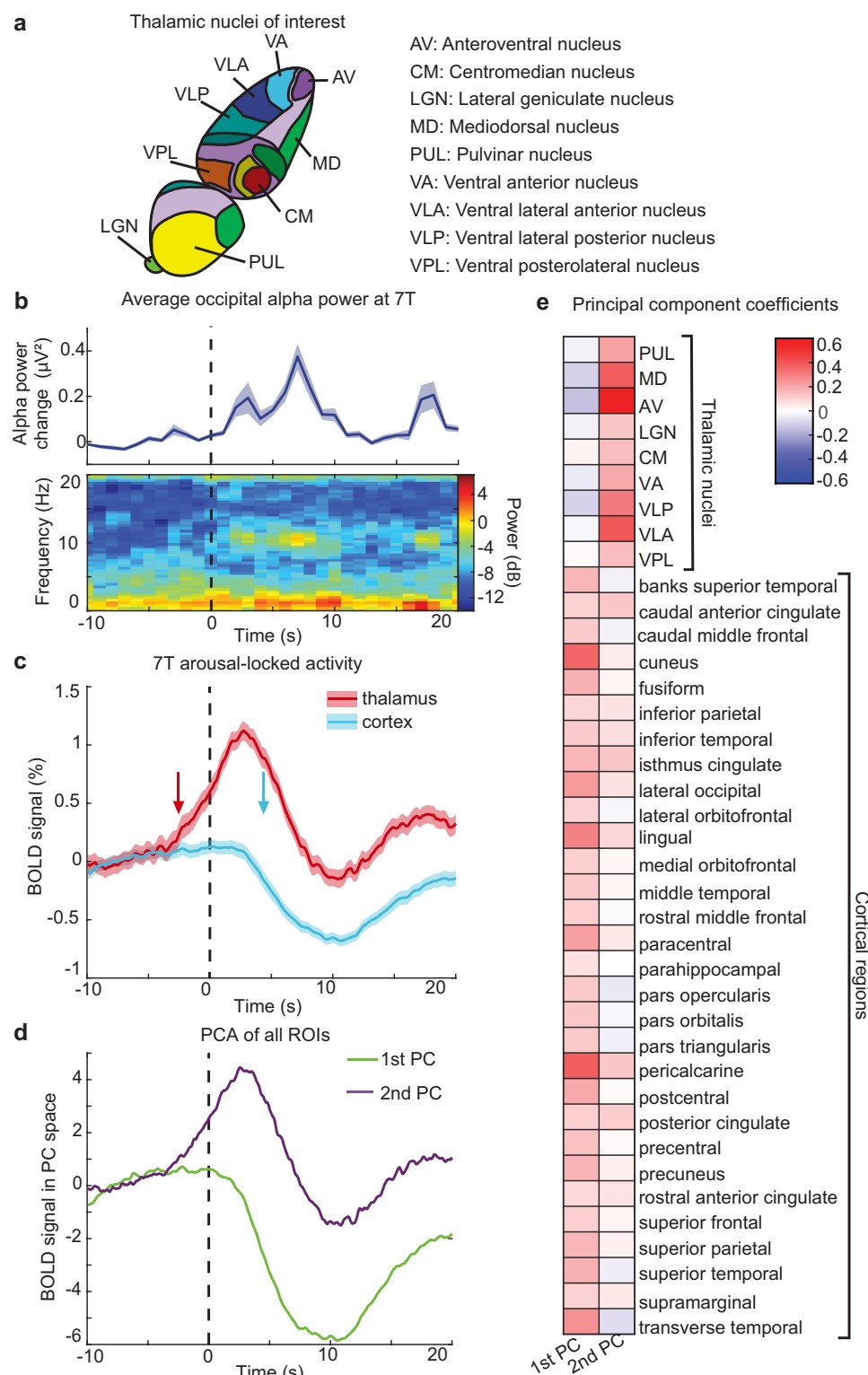

**Fig. 3 | 7 T imaging shows two primary activity modes at behavioral arousal, largely segregated into thalamic activation and cortical deactivation.**
**a** Schematic of the thalamic nuclei of interest that were resolved with this imaging protocol. This image was created by the authors and was inspired by the Allen Brain Atlas[131]. **b** In one subject scanned with simultaneous EEG ($n = 6$ arousals), mean alpha power significantly increases during behavioral arousal (dashed line), consistent with Experiment 1. Shading is a standard error. **c** Experiment 2 replicates the result that the thalamus activates (20% latency indicated by red arrow) prior to

behavioral arousal, while cortical deactivation (blue arrow) follows ($n = 13$ subjects, 99 arousals). Shading is standard error. **d** A principal component analysis of all nine thalamic regions and 30 cortical regions reveals two primary modes of activity in the thalamus and cortex: one which increases before arousal and one which decreases after arousal. **e** The first principal component is more heavily influenced by cortical regions, and the second is influenced primarily by thalamic regions, demonstrating segregation of thalamic and cortical activity during arousal. Source data are provided in the "Fig. 3 Source Data" file.

In one subject, we again confirmed an electrophysiological arousal state transition during behavioral arousal by simultaneously recording EEG signals[71] to measure occipital alpha. As at 3 T, we found that occipital alpha power rose during arousal ($n = 6$ arousals, Fig. 3b, $p = 0.01$ paired $t$-test), suggesting our behavioral definition captured similar arousal changes in the 7 T setting. We then repeated our analysis of the thalamocortical activity locked to arousal and replicated our observation that the thalamus activated beforehand (20% latency = −2.59 s, 95% CI:(−6.11 s, −0.93 s)) and the cortex deactivated afterward (20% latency = 4.38 s, 95% CI:(3.27 s, 5.56 s); Fig. 3c and Fig. S3). This relative timing was also consistently observed when using different methods for defining the baseline (Supplementary Fig. 3). Thus, these thalamic and cortical activity patterns were remarkably preserved across the two experiments, with a consistent rise in thalamic activity beginning before behavioral arousal transitions.

While our initial results identified a major divergence in overall cortical and thalamic activity at behavioral arousal, we next aimed to determine whether these dynamics were consistent within individual thalamic nuclei and cortical regions or unfolded heterogeneously across subregions. We found that all thalamic nuclei activated during behavioral arousal, and inter-nucleus correlation was higher during behavioral arousal than during wakefulness ($p < 0.001$, paired $t$-test; average $r = 0.31$ in wakefulness and $r = 0.46$ during arousal). In contrast, cACC and PCC were the only cortical regions that significantly activated during arousal, and all cortical regions deactivated afterward ($p < 0.05$, $t$-test Bonferroni corrected, Supplementary Fig. 4). To identify the primary modes of activity across regions in a data-driven way, we performed a principal component analysis to extract the principal components of all thalamic and cortical regions during arousal. We found that most (98.25%) variance could be explained with just two components (Fig. 3d). The first component was driven primarily by cortical regions and exhibited a large decrease after arousal, and the second component was driven primarily by thalamic regions, with activity increasing prior to arousal (Fig. 3e). These results demonstrated that thalamic and cortical regions were largely segregated during behavioral arousal, with the exception of distinctly thalamic-like dynamics in the cingulate cortex.

## An activity sequence across thalamic nuclei prior to behavioral arousal

While fMRI has previously identified spatial maps of regions implicated in arousal state transitions[51], we used the high temporal resolution of our imaging in Experiment 2 to test whether a temporal ordering of activity appeared across the thalamus during behavioral arousal. Specifically, does activity in one nucleus foreshadow subsequent thalamic-wide activation, or does the thalamus broadly activate as a whole? To compare the temporal properties of individual thalamic nuclei, we calculated the mean arousal-locked fMRI signal across time (Supplementary Fig. 5). To quantify relative timing across the thalamus, we implemented a cross-correlation analysis which calculated the lag between each thalamic nucleus and the mean signal from the whole thalamus. We found that the arousal-locked fMRI signal across thalamic nuclei exhibited highly heterogeneous timing properties, spanning a range of 2.1 s (Fig. 4a). A bootstrap analysis (Supplementary Fig. 6) demonstrated that the centromedian (CM) and ventral posterolateral (VPL) nuclei consistently led the rest of the thalamus, whereas the ventral lateral anterior (VLA), ventral anterior (VA), and anteroventral (AV) nuclei lagged (Fig. 4a). This sequence was robust to cross-subject variance, as it was conserved when using a hierarchical bootstrap across subjects (Supplementary Figs. 7 and 8a) as well as when excluding the two subjects with the most arousals (Supplementary Fig. 8b). Thus, a subset of spatially distinct thalamic nuclei consistently showed early activity, with subsequent activity unfolding across the remaining nuclei during arousal (Fig. 4b).

We next investigated whether this thalamic heterogeneity reflected a temporal sequence specifically in the onset time of activity across nuclei, since onset time may be more reliable than peak amplitude time in determining lags between regions by using fMRI[63]. We fit a double-Gaussian model to estimate a smooth hemodynamic response function (HRF) for each thalamic ROI, which allowed us to define onset time (the time when HRF reaches 10% of maximum amplitude, Fig. 4c and Supplementary Fig. 9). We found that CM indeed activated first, 2.56 s before the global thalamus, which had an onset time of −5.97 s. CM was 0.68 s earlier than the next fastest nucleus, the VPL, and 4.08 s before the last nucleus, VA (Fig. 4c, d). The relative timing of the temporal sequence of thalamic activity was thus largely conserved across methods. This result confirmed a specific temporal sequence of thalamic activity underlies the transition between behavioral arousal states, with CM activity appearing first, VPL next, and these earliest nuclei activating seconds prior to the rest of the thalamus.

While thalamic regions activated before arousal, another primary signature of behavioral arousal-locked dynamics was widespread cortical deactivation. To determine if there was a sequence of deactivation across the cortex, we next investigated the spatiotemporal properties across the cortex. We calculated the lag between each cortical region and the whole cortex and found a range of relative deactivation times (Supplementary Fig. 10a). Early regions included several parietal and frontal regions, whereas late regions included all of the cingulate cortex and inferior temporal cortex (Supplementary Fig. 10b). These timing patterns shared some similarities with prior results showing that a traveling wave propagates across the cortex during fluctuations in arousal level[48], suggesting that intermittent arousals could contribute to ongoing spatiotemporal dynamics across the cortex as well.

## Vascular physiology cannot explain thalamic latency differences

Since blood oxygenation level-dependent (BOLD) fMRI signals arise from hemodynamic responses to neural activity, and this hemodynamic response can be heterogeneous across the brain[60,72–75], we next examined whether differences in vascular reactivity existed across thalamic nuclei that could contribute to these temporal sequences in Experiment 3. As a control experiment, we measured fMRI signals caused by the widespread vascular response to a brief breathhold in order to assess hemodynamic latency differences across nuclei[76–82]. We found that the breathhold task produced robust fMRI signals across thalamic nuclei (Supplementary Fig. 11a). The amplitude of breath-locked responses across thalamic nuclei was strongly correlated with the previously observed arousal-locked signal amplitudes ($r = 0.88$; Supplementary Fig. 11b), demonstrating that the breathhold task provided a reliable control experiment to investigate hemodynamic properties of thalamic nuclei. Therefore, we repeated our temporal sequence analysis in these control data. We found that the thalamus led the cortex by only −0.12 s (95% CI: (−0.43 s, −0.062 s)) during the breathhold task (Fig. 5a), far less than the 6.42 s (Experiment 1) and 6.97 s (Experiment 2) latency differences observed at arousal, and consistent with prior studies showing slightly faster hemodynamics in thalamus than in cortex[61,83,84]. Furthermore, while minor differences in the breathhold fMRI response were detected across thalamic nuclei, the lag differences between nuclei could not account for the differences observed during arousal (Fig. 5b). One nucleus, the AV, lagged significantly behind the whole thalamus by 0.99 s (CI = (0.12 s,1.6 s)), suggesting that the late arousal-locked activation of AV was partly due to a slower hemodynamic response rather than delayed neural activation. In addition, the mediodorsal (MD) nucleus had slightly slower, the VPL and VLA nuclei exhibited slightly faster (less than 0.25 s in each nucleus), and the CM, pulvinar (PUL), lateral geniculate (LGN), ventral lateral posterior (VLP), and VA nuclei showed similar vascular reactivity to the whole thalamus. Altogether, these differences across nuclei were small, and could not explain the multi-second thalamic sequence observed at behavioral arousals. Furthermore, we corrected our

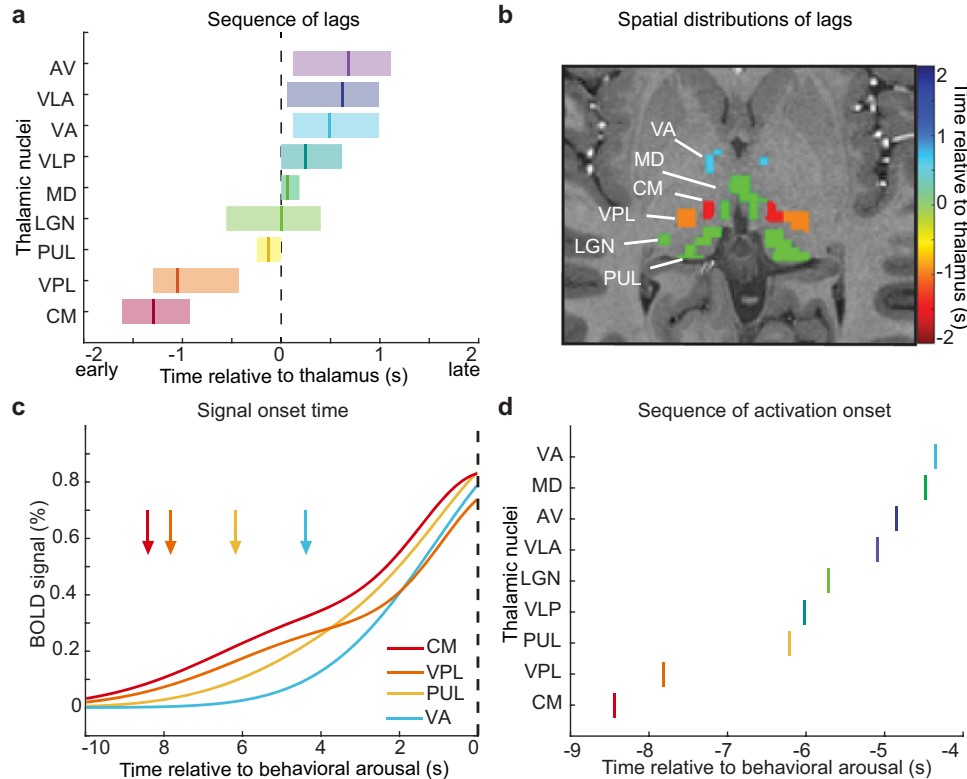

**Fig. 4 | A sequence of activity occurs across thalamic nuclei during behavioral arousal. a** The relative timing of thalamic activation differs significantly across nuclei (mean lag shown by the solid vertical line; shading is 95% confidence interval, colored from red to purple based on the order in sequence). Zero represents the timing of the whole thalamus (dashed line). The centromedian (CM, red) and ventral posterolateral (VPL, orange) thalamic nuclei lead the rest of the thalamus during behavioral arousal, while the ventral anterior (VA, light blue), anteroventral

(AV, dark blue), and ventral lateral anterior (VLA, purple) lag behind. **b** Image of mean lag times per nucleus in one subject. Color represents precise lag. **c** The activation onset times (arrows) and the hemodynamic response (solid lines) show a sequence in activation across nuclei. Color represents the order in a lag sequence. **d** Thalamic nuclei activation onset is sequenced similarly to the lag values. Source data are provided in the "Fig. 4 Source Data" file.

arousal sequence for each nucleus' hemodynamic latency and still found equivalent groupings of early and late nuclei (Supplementary Fig. 11c). Similarly, vascular differences could not explain the temporal dynamics across the cortex (Supplementary Fig. 11d). These results supported the conclusion that a temporal sequence of neural activity across thalamic nuclei occurs prior to the moment of behavioral arousal, with the earliest activation first in CM and then in VPL.

A second modulator of fMRI signals is systemic physiology: systemic physiology changes across arousal states, and in particular, changes in heart rate and respiration accompany arousal[85–87]. Such shifts in systemic physiology can also contribute to the fMRI signal[76,88–91]. To investigate how these systemic physiological dynamics were linked to behavioral arousal, we calculated the average respiratory and fingertip piezoelectric pulse amplitude using the simultaneously recorded physiology in Experiment 2. We indeed found that changes in the respiratory (Fig. 5c) and pulse (Fig. 5d) signal amplitudes were locked to behavioral arousal, consistent with known physiological modulation linked to arousal. However, both the respiratory and pulse signals changed after behavioral arousal. These systemic physiological effects, therefore could not produce hemodynamic confounds to explain the thalamic activity beginning seconds earlier. Altogether, these results demonstrated that shifts in systemic physiology accompany the transition in behavioral state, consistent with the expected physiological changes during shifts in arousal, but that they occurred after the arousal-locked thalamic activity sequence began. We thus concluded that these thalamic temporal dynamics observed in Experiment 2 represented a neural activity sequence that began to unfold across thalamic nuclei before the moment of behavioral arousal.

## Thalamic signals differ in sustained and transient behavioral arousals

Behavioral dynamics after arousal can vary substantially: the subject may fully transition to the alert state and maintain active behavior, or may fall directly back into unresponsiveness after only brief arousal. Our study thus provided an opportunity to ask whether these thalamic dynamics reflected simply the moment of transition between states, or also the maintenance of a subsequent higher arousal state. We separated the behavioral arousals into two types (Fig. 6a): sustained arousals (at least five responses following arousal) and transient arousals (two or fewer responses), resulting in 45 sustained arousals and 34 transient arousals. Surprisingly, even in brief arousals consisting of two or fewer button-presses, we found strong activity increases in the thalamus (Fig. 6b). However, the thalamic activity plateaued at a level higher than its pre-arousal baseline during sustained arousals, whereas after transient arousal, the thalamus deactivated and returned to baseline (Fig. 6b). Interestingly, the mean cortical signal did not have a notable difference between fMRI signatures in transient versus sustained arousals (Fig. 6c). Therefore, a stable plateau of thalamic activity reflected differences in subsequent maintenance of arousal.

We next evaluated dynamics across individual thalamic nuclei and cortical regions in sustained and transient arousals to test whether they were linked to the subsequent maintenance of arousal. We found that most thalamic nuclei exhibited higher post-arousal plateau activity in sustained arousals, similar to the whole thalamus (Supplementary Fig. 12). The LGN demonstrated the smallest difference between sustained and transient arousals, whereas all other thalamic nuclei showed significant differences ($p < 0.05$ paired t-test Bonferroni corrected, Supplementary Fig. 10). In contrast, across cortical regions,

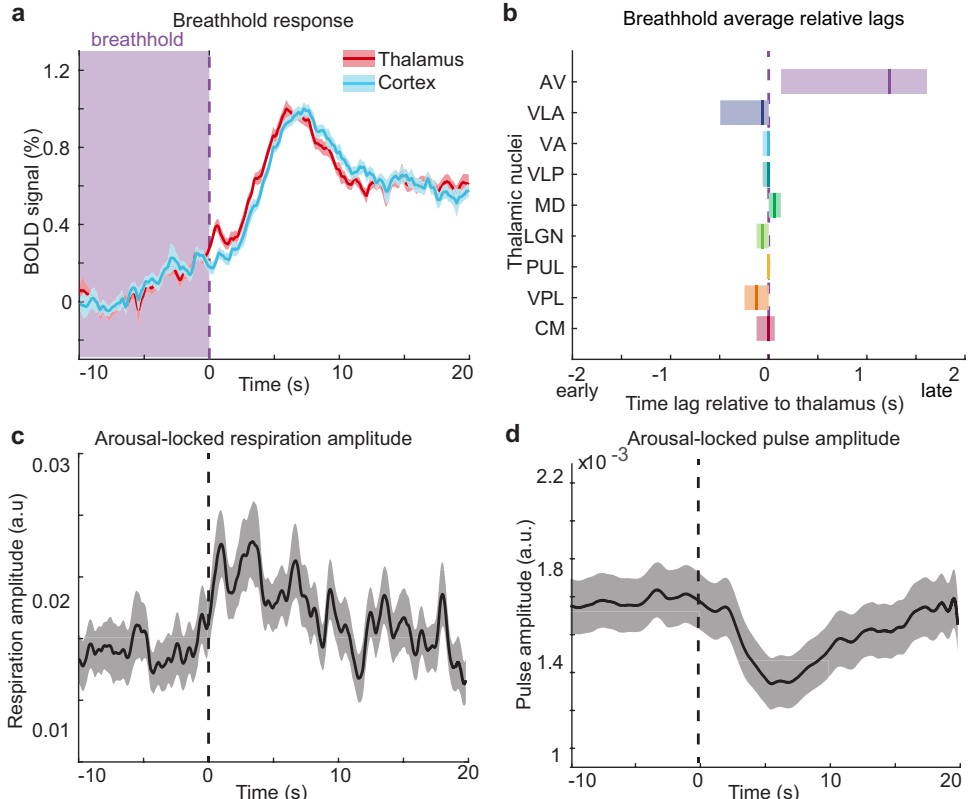

**Fig. 5 | Hemodynamic latencies cannot explain the arousal-locked temporal dynamics in the thalamus. a** fMRI signals increase after breathhold release (purple dashed line). The time of the breathhold is shaded in light purple. The thalamus leads the cortex by 0.12 s. Data were presented as mean values, and shading represents standard error. **b** Small hemodynamic lags between thalamic nuclei exist but are not large enough to account for the arousal-locked sequence. The color and ordering of thalamic nuclei represent the lag sequence during arousal.

Mean lag (solid line) and 95% confidence interval (shading) were calculated by the same method as Fig. 4d. Purple dashed line represents a zero-lag relative to the whole thalamus. **c** Respiration amplitude increases at behavioral arousal (dashed line). Data were presented as mean values, and shading is the standard error. **d** Amplitude of the pulse signal decreases after behavioral arousal. Shading is a standard error. Source data are provided in the "Fig. 5 Source Data" file.

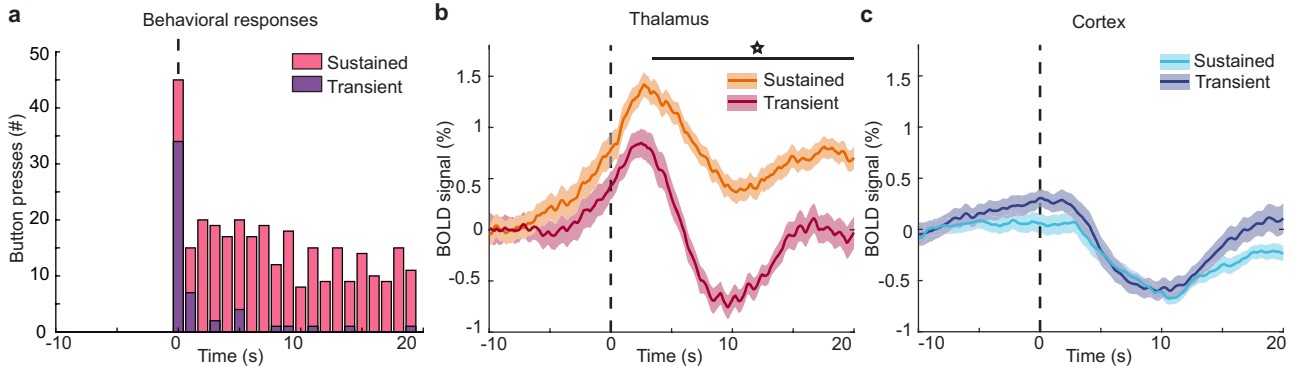

**Fig. 6 | The late plateau of thalamic activity differs across transient and sustained arousals. a** During sustained arousals (pink), the behavior continues after the moment of arousal, whereas in transient arousals (purple), participants lapse back into unresponsiveness. **b** The post-arousal plateau in thalamic activity is significantly higher (starred, black line, Bonferroni corrected paired *t*-test, $p < 0.05$) in

sustained arousals. Data were presented as mean values, and shading represents standard error. **c** In the global cortex, there is no significant difference in sustained versus transient arousals. Shading is a standard error. (Individual cortical ROIs are in Supplementary Fig. 11.) Source data are provided in the "Fig. 6 Source Data" file.

only the parahippocampal cortex showed a significant difference in sustained and transient arousals (Supplementary Fig. 13). Next, to determine if the temporal sequence differed across arousal types, we repeated our cross-correlation lag analysis. We found that in both sustained and transient arousals, there was a significant activity sequence across thalamic nuclei, beginning with the CM (Fig. 7). However, thalamic nuclei activated more closely together in time

during sustained arousals, with a range of 1.66 s (Supplementary Fig. 7a). During transient arousals, the activity sequence was longer, with a range of 2.47 s (Fig. 7b). Furthermore, while CM was early during both types of arousals, VPL was early only in the transient arousals. These results were robust to intersubject variability (Supplementary Fig. 14). Overall, during sustained switches in behavioral state, the temporal activity sequence across nuclei concluded in coordinated

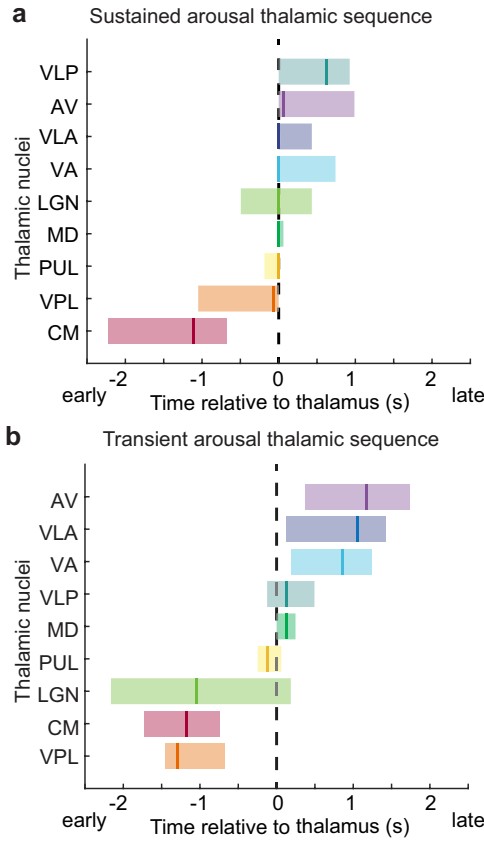

**a** Sustained arousal thalamic sequence

**b** Transient arousal thalamic sequence

**Fig. 7 | Sequence of thalamic activity in sustained and transient arousals. a** The sequence at sustained arousals demonstrated that CM was still the first to activate, and the subsequent sequence across nuclei occurs closely together in time. The vertical line is the mean lag, shading is a 95% confidence interval. **b** The sequence at transient arousals evolved more slowly across nuclei. Source data are provided in the "Fig. 7 Source Data" file.

activity throughout most of the thalamus, whereas transient arousal was accompanied by a more sluggish sequence across nuclei. Thus, network activity across thalamic nuclei reflected whether the transition in behavioral arousal state was subsequently maintained.

## Discussion

We conclude that behavioral arousal is preceded by a temporal sequence of activity across thalamic nuclei and cingulate cortex, followed by widespread deactivation throughout the thalamus and cortex. A consistent temporal pattern emerged across individual thalamic nuclei at the moment of state transitions, and this thalamic activity was linked to subsequent maintenance of behavioral arousal. Our results are consistent with prior animal[18–22] and human fMRI studies[50,51,68] showing that thalamic activity is coupled to arousal state. Our work further demonstrates that thalamocortical networks exhibit highly structured spatiotemporal dynamics that evolve throughout this transformation between unresponsiveness and active behavior, with distinct profiles for specific thalamic nuclei.

We consistently found that CM, an intralaminar nucleus that receives input from the reticular activating system as well as sensorimotor information[92,93], activated first within this thalamic temporal sequence. This is consistent with studies in mice, which found that during NREM to wake transitions, the central medial nucleus (CeM), another intralaminar thalamic nucleus, activates earlier than the ventrobasal nucleus, a sensory thalamic nucleus, and optogenetically driving CeM induces awakening[34]. Furthermore, electrical stimulation of the central thalamus causes behavioral arousal transitions in

anesthetized macaques[32,39], and deep brain stimulation of the central thalamus modulates behavioral responses in minimally conscious state patients[94]. In line with these results, we demonstrate that in spontaneous behavioral arousals, activity originates in an intralaminar nucleus and subsequently spreads to other thalamic nuclei. While the involvement of intralaminar nuclei in arousal modulation is well supported by prior work, our study also identified VPL as a second thalamic nucleus which consistently activated early. This finding was unexpected, as VPL is a first-order somatosensory nucleus[95] which has not been previously implicated in arousal-state changes, and suggests that the sensory thalamus may contribute to state transitions as well. Interestingly, VPL was early only in the transient arousals, suggesting that brief arousals are more likely to have an early contribution from the sensory thalamus.

Our results suggest potential mechanisms that could generate such a thalamic temporal sequence. One possibility is that the sequence of activity across thalamic nuclei is the result of a traveling wave which propagates across the cortex and subcortical regions during ongoing fluctuations in arousal[48,49]. This sequence could also be caused by activity spreading across the thalamus. Arousal-driving intralaminar thalamic nuclei may shift baseline excitation across the thalamus via the thalamic reticular nucleus or cortex, setting the stage for subsequent thalamic activation and arousal. If only a subset of nuclei, such as the intralaminar nuclei, can effectively raise baseline excitation in neighboring nuclei and induce broader thalamic synchronization, this may explain why stimulating subcomponents of the chain does not cause arousal transitions[34,39]. Alternatively, slow inputs from the ascending arousal system may cause sequential activation across the thalamus during arousal, perhaps due to differences in local neuromodulatory release or differences in neuromodulatory receptor density across nuclei. CM and VPL could receive stronger neuromodulatory input, causing them to activate earlier, while activity in other thalamic nuclei could then follow due to later/weaker input. Furthermore, our results suggest that more coordinated activation across nuclei may contribute to more effective transitions in the arousal state; this finding could reflect more potent neuromodulatory input from brainstem arousal nuclei that results in a sustained behavioral state change.

We found that while activity in most of the cortex declined at arousal, the ACC and PCC notably diverged and instead mirrored the thalamus with a large increase in activity before arousal. This suggests that the cingulate cortex could also contribute to behavioral arousal state transitions. Prior studies have shown that these regions are linked to state-switching and attentional modulation within the awake state: the ACC is involved in sustained arousal in complex, novel environments[96] and attentional-state switching[97], and the dorsal ACC is active during moments of surprise[98]. The PCC supports internally directed cognition[99], regulation of the focus of attention[100,101], and is a component of the default mode network, which is more active in the task-negative state[102,103]. Our results thus align with the role of the cingulate cortex in attentional-state switching and further suggest that these cingulate regions may also have a broad, non-specific role in task engagement.

Our finding that both cortex and thalamus deactivated after the moment of behavioral arousal raises several possible interpretations, as both neural mechanisms as well as indirect neuromodulation of vasculature could potentially contribute to this decrease in BOLD signal[104]. Neural inhibition could cause a decrease in BOLD[105]; however, the spike rate of most neurons in the cortex increases slowly after awakening[106], suggesting the decline in cortical signal may have different origins. One contributing factor could be the shift from coherent to desynchronized cortical activity that occurs with arousal, due to the change from burst to tonic firing in thalamus[107]. This desynchronization may drive decreased fMRI signals, since changes in neural synchrony modulate the BOLD signal[108]. Additionally, the decrease in

cortical BOLD signal could also reflect vascular factors, as it is possible for cerebral blood flow to decrease despite increased neural activity[109]. This could be caused by a redistribution of oxygenated blood to more active areas, such as deep brain regions[110], or endogenous alterations in vasoconstriction due to neuromodulators such as noradrenaline[111,112]. Notably, this cortical decrease may also be partly a result of systemic vascular changes, as arousal is also often followed by systemic vasoconstriction[88,90,91]. Therefore, a combination of hemodynamic and neural mechanisms may thus contribute to the decrease in BOLD signal after arousal.

Since vascular dynamics can influence the fMRI signal and are strongly modulated by the arousal state, we performed control analyses to confirm that the activity sequence observed across thalamic nuclei was not driven by hemodynamic confounds. Our breathhold experiment tested for local differences in cerebrovascular reactivity, and while the hemodynamic response to a myogenic breathhold task may not exactly replicate the neurogenic vascular response[79], it can be used to correct for hemodynamic lags across the brain[77]. Correcting for hemodynamic lags demonstrated that most thalamic nuclei had only very small differences in vascular reactivity, with the exception of a notably slow response in AV. Exactly why AV has a slower hemodynamic lag is not clear, but could reflect differences in its vascular architecture[73]. In addition, we found that systemic physiological changes in the pulse and respiration occur after arousal, and, therefore could not explain the early thalamic activation prior to arousal. Our observations of temporally patterned thalamocortical dynamics were therefore robust even after accounting for vascular physiology.

Prior fMRI studies have reported an overall thalamic activation and cortical deactivation at behavioral arousal[51], but with important differences. Specifically, our study demonstrated that the thalamus activates long before the cortex deactivates, replicated in two separate experiments. Our detection of precise temporal sequences can be primarily attributed to the fast temporal sampling of our acquisition, which enhances sensitivity to timing differences[113,114]. However, a second potential contributor to this difference is that prior work aimed to reduce physiological noise by regressing the cerebrospinal fluid (CSF) signal, a common approach in fMRI. However, CSF signals are strongly coupled to neural activity and the fMRI signal during sleep[67]. Regressing out the CSF could therefore remove neurally relevant dynamics and distort the timing of the underlying signals[115]. This issue reflects a broader challenge in neuroimaging studies of sleep and arousal: the neural dynamics that drive arousal state changes are often collinear with the systemic physiological changes they induce, and common noise removal practices can have unintended effects.

This study focused on behavioral arousal state transitions, which can often be distinct from the classic electrophysiological arousals from sleep. Importantly, thalamocortical network activity may differ during electrophysiological arousals—for example, our results would likely be different if examining arousal defined by the return of occipital alpha in the EEG. In that scenario, we would predict that the LGN might deactivate during the return of occipital alpha rather than activate as we report in behavioral arousals, due to the established anticorrelation between LGN activity and alpha rhythms[116]. Future studies could use this accelerated imaging approach to delineate the thalamic activity linked to electrophysiological, as compared to behavioral arousal transitions.

Along these lines, future work could also investigate distinct types of arousal transitions. Here, we studied spontaneous behavioral arousal, but there are many ways to awaken from sleep and different sleep stages from which arousal can occur. An intracranial study in humans found that arousal-locked thalamic spectral signatures are highly stereotyped across different sleep stages and in stimulus-driven arousals[117]. However, the precise timing of these activity patterns has not yet been measured, and the order of activation across thalamic nuclei could be quite different. For example, one might expect the visual nuclei of the thalamus (LGN and PUL) to activate earlier during arousal driven by visual stimulation, such as the lights turning on in a darkened room.

In conclusion, we identified a sequence of activity that unfolds across thalamic nuclei during behavioral arousal state transitions. Our results reinforce the role of the thalamus in arousal state transitions and identify distinct functional profiles for the diverse nuclei within the thalamus, uncovering precise temporal patterns that are coupled to both the moment of transition and the stability of the subsequent arousal state. Finally, these results suggest a broad potential for fast, ultra-high field fMRI to identify the temporal dynamics of subcortical activity in the human brain. As subcortical structures are challenging regions to image but play fundamental roles in cognition and awareness, this highlights a valuable approach for studying their functional roles at sub-second timescales.

## Methods

### Participants

The current study includes three datasets. All experimental procedures were approved by the Massachusetts General Hospital Institutional Review Board. Participants in all datasets were screened not to have any neurological, psychiatric, or sleep disorders and not currently taking psychiatric or sleep medications. Subjects in all three datasets provided written informed consent. The first dataset (Experiment 1) used 3 T fMRI data from a previously published study of cerebrospinal fluid dynamics during sleep[65]. Our current analysis of these data included the subset of subjects who were instructed to perform a behavioral task during the sleep imaging ($n = 6$; one male and five female), with a mean age of 24.6 years (range: 23–26).

The second dataset (Experiment 2) was a newly acquired imaging experiment using the same behavioral task, performed with fast fMRI at 7 T. Written informed consent was obtained from 20 healthy adults (14 female and six male; mean age: 24.9, age range: 22–33). Seven of these subjects were scanned with simultaneous EEG. Participants were excluded from all analyses if they lacked any behavioral arousals ($n = 5$) or excessive motion artifacts ($n = 2$). In total, 13 subjects were included in the analysis (nine females and four males, mean age = 25.2, age range: 22–33), four of whom had simultaneous EEG. Subjects who did not present occipital alpha were excluded from the EEG analysis portion only, which resulted in one subject with simultaneous 7 T fMRI and EEG being included in our EEG analysis.

For the third dataset (Experiment 3), eight subjects participated in the 7 T fMRI breathhold control experiment (three females and five males, mean age = 25.5, range: 20–34), with the same exclusion criteria as Experiment 2. Three of these individuals were also participants in Experiment 2. No participants were excluded.

### Sleep and behavioral protocols—Experiments 1 and 2

Participants were asked to restrict their sleep to four hours the night prior to the sleep scan (Experiments 1 and 2) and avoid caffeine 24 h prior to the study. Subjects self-reported the number of hours they slept. Imaging sessions began between 10:30 p.m. and midnight, the night after sleep restriction.

Participants in Experiments 1 and 2 were instructed to keep their eyes closed for the scan duration. Participants were instructed to press a button on every breath in and breath out as long as they were awake and that it was permitted to fall asleep during the scan. Behavioral arousals were defined as the first response to this behavioral task after at least 20 s of unresponsiveness. Behavioral arousals that included excessive motion (>0.3 mm) were excluded from the analysis.

### BOLD fMRI acquisition

Participants in Experiment 1 lay supine in an MRI scanner (Siemens 3 T Prisma, 64-channel head-and-neck coil). First, we performed a $T_1$-weighted anatomic multi-echo MPRAGE scan with 1-mm³ isotropic

voxels to provide an anatomical reference[118]. Participants then underwent BOLD-weighted echo-planar-imaging (EPI) scans for fMRI data acquisition covering most of the brain. The acquisition volume was positioned consistently across subjects at an oblique angle, tilted to avoid capturing signals from the eyes to minimize eye motion artifact. This positioning captured most of the brain but omitted the tip of the temporal lobe and the lower half of the cerebellum. This functional scan contained up to 1500 volumes and lasted up to 90 min per run. The run typically lasted 90 min if no other factors intervened but was sometimes shorter and followed by a subsequent run if scanning needed to be paused, for example, if the subject pressed the squeeze ball or if a sensor needed to be adjusted. Single-shot gradient-echo SMS-EPI[57] data were acquired with a voxel size of 2.5 mm³ isotropic, MultiBand factor = 8, matrix = 92 × 92, shift factor = 4, TR = 367 ms, TE = 30 ms, flip angle = 32–37°, echo spacing = 0.53 ms, and no in-plane acceleration.

Participants in Experiment 2 lay supine in an MRI scanner (Siemens 7 T Magnetom, custom-built 32-channel head coil). First, a $T_1$-weighted anatomic multi-echo MPRAGE scan with 0.75 mm³ isotropic voxels was performed to provide an anatomical reference[118,119]. Participants then underwent one to three EPI scans for fMRI data acquisition, containing up to 8000 volumes and lasting up to 33 min. Subjects performed the same self-paced behavioral task as in Experiment 1. Since the sequence used only allowed for up to 33-min duration scans, we targeted three scans per subject to acquire the same amount of data as in Experiment 1. Similarly, the scans were sometimes ended early if needed for technical reasons, for example, if the subject accidentally pressed the squeeze ball while sleeping, as in Experiment 1. The acquisition volume was positioned consistently across subjects, aligning the bottom edge with the lower entrance to the fourth ventricle and tilting it upward to avoid capturing signals from the eyes. This positioning did not include the tip of the temporal lobe and the lower half of the cerebellum. 40 oblique slices were acquired with a voxel size of 2.5 mm³ isotropic, TE = 24 ms, TR = 247 ms, MultiBand factor = 8, shift factor = 4 matrix = 84 × 84, flip angle = 30°, echo spacing = 0.53 ms, and no in-plane acceleration.

Participants in Experiment 3 underwent the same imaging protocol as Experiment 2, but with functional runs split into 8-min periods. Each participant underwent four to eight functional runs where they performed the breathhold task.

Physiological signals were simultaneously recorded during BOLD acquisition. In Experiment 1, physiological signals were recorded using a Physio16 device (Electrical Geodesics, Inc., Eugene, OR, USA). ECG was measured through two disposable electrodes placed on the chest diagonally across the heart with an MR-compatible lead (InVivo Corp, Philips). Respiration was measured through a piezo-electrical belt (UFI systems, Morro Bay, CA, USA) around the chest. In Experiments 2 and 3, pulse was measured with a piezoelectric pulse transducer around the left thumb (AD Instruments, Colorado Springs, CO, USA). Respiration was measured through a piezo-electrical respiration transducer belt around the chest (UFI, Morro Bay, CA, USA). Both of these were acquired using LabChart version 7.

## fMRI preprocessing

Preprocessing of MRI data was performed using Freesurfer developmental version from 08-12-2019 (https://surfer.nmr.mgh.harvard.edu/fswiki) and FSL version 5[120] (https://fsl.fmrib.ox.ac.uk/fsl/fslwiki). Anatomical images were bias-corrected using SPM version 12 in MATLAB and then automatically segmented using Freesurfer[121]. Functional images were realigned to correct motion artifacts using AFNI version 19.1 (https://afni.nimh.nih.gov/), and slice-time corrected using FSL version 5.

Fast fMRI is susceptible to physiological noise[89,122], including cardiac rhythms and respiration, which appear in the data as quasiperiodic BOLD oscillations. We used dynamic regression based on the concept of RETROICOR[123] to remove signals driven by the heartbeat and respiration from the data while allowing the peak frequency of these physiological rhythms to vary over time. The cardiac signal was bandpass filtered between 0.2 and 10 Hz. Peaks of the cardiac signal were identified using the automated peak detection technique in the Chronux toolbox version 2.12 (http://chronux.org/)[124], and the inter-peak intervals were transformed into phases. The respiratory signal was bandpass filtered between 0.16–0.4 Hz using a finite impulse response filter, and the instantaneous phase was computed as the angle of the Hilbert transform. This phase information was transformed into sine functions, and beta values were estimated in a window of 1000 s sliding every 400 s voxel-wise using a general linear model. These values were then interpolated across each time-point and used to remove each voxel's first and second harmonic frequencies of the cardiac and respiratory signals.

Spatial smoothing was not applied to avoid blurring and to maximize spatial accuracy within the thalamus. All fMRI signal analyses were performed in the original spatial frame of the fMRI acquisition space for each individual without transforming to a common average. Registration between the functional and anatomical images was completed using Freesurfer boundary-based registration[125]. Cortical ROIs were extracted using the Desikan–Killiany atlas[69] to identify functional voxels that were at least 70% filled by each region. The thalamic segmentation was done using the individual-level probabilistic atlas in the Freesurfer developmental version, which provides voxel-wise segmentation probabilities in individual anatomical space, that have been previously validated against ex vivo histology[70]. Nuclei were defined by selecting functional voxels that had at least a 90% chance of falling within a given thalamic nucleus to minimize partial-volume effects. All voxels that fell in any of the four pulvinar subregions were combined into a single pulvinar nucleus ROI, and similarly, any voxels than fell in either of the two mediodorsal subregions were combined into the mediodorsal nucleus ROI. Cortical and thalamic regions which were not captured in every subject were excluded from the analysis. The mean time course in each cortical and thalamic ROI was then extracted using FSL version 5.

## EEG recordings, preprocessing, and analysis

In Experiment 1, EEG was measured by an MRI-compatible 256-channel geodesic net and a NA410 amplifier (Electrical Geodesics, Inc., Eugene, OR USA) at a sampling rate of 1000 Hz using EGI acquisition software version 5.4. Each subject wore a reference layer cap composed of an isolating vinyl layer and conductive satin layer on the head, with grommets inserted to allow electrodes to pass through and make contact with the scalp[126], while other electrodes remained isolated from the scalp and recorded the noise, resulting in a total of 30–36 EEG electrodes per subject. The MR scanner's helium pump was temporarily shut off during EEG acquisition to reduce vibrational artifacts in the EEG signal. In a subset of the analyzed subjects ($n = 4$) in Experiment 2, EEG was measured by a specially designed 256-channel polymer thick film MR-compatible EEG net, called the InkNet, for EEG collection at 7T[71]. A custom reference cap was placed under the InkNet to measure ballistocardiogram (BCG) noise[67,126]. EEG acquisition was synchronized to the scanner 10 MHz master clock to reduce the aliasing of high-frequency gradient and RF-induced artifacts. A baseline of eyes open versus eyes closed, eye movement, and jaw clenching were recorded outside of the scanner.

EEG event data were loaded into MATLAB using Fieldtrip version 20191025. Gradient artifacts were cleaned from the EEG data using average artifact subtraction[127]. EEG data were filtered using a linear-phase filter between 0–50 Hz and then down-sampled to 200 Hz. The EEG channels and BCG channels were re-referenced to their respective average. All channels on the cheeks were excluded from this average. The BCG artifact was then removed by adaptively regressing the

influence of the BCG channels' signal from each EEG channel over a sliding window of 30 s[126–128].

EEG was used to determine the sleep stage prior to behavioral arousal and if these behavioral arousals correlated with a shift in electrophysiological arousal. The EEG data in Experiment 1 were sleep-scored in 30-s intervals using the standard vigilance state scoring criteria defined by the AASM[2]. The sleep stage in the 30-s window 15 s prior to arousal was recorded.

The spectrogram of the cleaned EEG data was computed to identify shifts in electrophysiological arousal during behavioral arousals. The alpha power (7–12 Hz) was calculated in the electrode closest to the Oz position. Power in the alpha frequency range was calculated using Chronux version 2.12[129] (tapers = 3, sliding window length = 2, sliding step = 1]). To test for a significant rise in alpha power during behavioral arousal, we compared the average occipital alpha power in the 10 s before behavioral arousal versus the 10 s after. We averaged the occipital alpha power in these time windows for each arousal and calculated if there was a significant change in alpha power using a two-sided, paired $t$-test. A $p$ value of less than 0.05 was considered significant.

### Behavioral arousal-locked signal analysis

After extracting the ROI timeseries, we linearly interpolated the fMRI signal to four times its original sampling rate (61.75 ms) in order to align the fMRI data more precisely with the moment of arousal and identified the sample closest to the arousal. We extracted the fMRI signal 10 s before and 20 s after this sample. Arousal windows that had a framewise displacement at any point over 0.3 mm were excluded from analyses. The resulting signals with low motion were averaged to calculate the mean fMRI response during arousal for each ROI. These means were centered at the average value during the first 3 s of the timeseries (−10 and −7 s). Standard error across arousals was defined as the standard deviation divided by the square root of the number of arousals.

To determine the latency of the thalamus and cortex, we identified the time at which the fMRI signal reached 20% of its maximum absolute amplitude. The baseline of the mean fMRI signal was calculated as the mean BOLD signal in the −10 to −7 s before arousal. We used a bootstrap analysis to calculate the 95% confidence interval of the thalamic and cortical latencies. First, we resampled the arousals with replacement to re-calculate the mean fMRI time course and latency of each region. Then, we recomputed the latency of the activity onset. We did this 1000 times. The 95% confidence interval was calculated by using the 2.5 and 97.5 percentiles. The 20% latency of the ACC and PCC were calculated based on their maximum positive amplitudes rather than their maximum absolute amplitudes to ensure the latency time was in relation to their activation and not deactivation.

To test for a significant change in any cortical region's fMRI signal during behavioral arousal, we used a time-binned, two-sided t-test. We averaged each ROIs fMRI signal across 1-s bins. Each bin was tested using a two-sided $t$-test to determine if the fMRI signal was significantly different from zero. The $p$ value was Bonferroni corrected for the number of time bins and cortical regions tested, and time bins with corrected $p$ values less than an alpha value of 0.05 were considered significant.

A principal component analysis was used to determine the major activation patterns that underlie behavioral arousal state transitions in all of the cortical and thalamic ROIs. The mean timeseries of each ROI 10 s before and 20 s after arousal was centered to the first 3 s of the timeseries (−10 and −7 s), and singular value decomposition was used. The number of components was the total number of inputs, 39.

The estimation of lags between ROIs during arousal was possible due to the high temporal resolution of our fMRI data. A cross-correlation analysis was used to compute the time lag that produced the maximal correlation between the mean signals of each thalamic nucleus and the whole thalamus, or each cortical region and the whole cortex. A bootstrap analysis was used to resample with replacement from arousals, generate a resampled mean timeseries, and calculate 95% confidence intervals of the lags between regions, resampling 1000 times. In the hierarchical bootstrap analysis, we resampled the subjects with replacement and then resampled arousals with replacement from those individual subjects, until the total number of behavioral arousals across resampled subjects was at least 97 (matching the true total number of arousals). After bootstrap resampling, the mean fMRI signal during arousal for each ROI was computed, and the cross-correlation analysis was repeated. The upper and lower bounds of the 95% confidence interval of the lag between each nucleus and the whole thalamus were computed by calculating the 2.5 and 97.5 percentiles.

Next, we determined the onset time of each thalamic ROI. Since these small, deep regions can be noisy, we fit a linear combination of two Gaussian curves to each ROI using a simple search method[130] to find the parameters with the smallest root mean squared error. We defined onset time as the time that the model fit reached 10% of its maximum amplitude. To allow for temporal flexibility, we included the temporal derivative of each Gaussian in the model fit.

To determine if thalamic and cortical activity deviated between arousals in which the subject continued behaving after the initial button-press or fell directly back into unresponsiveness, we segregated the behavioral arousals into arousals with five or more button-presses (sustained arousals) and two or fewer button-presses (transient arousals). Significant differences between fMRI signals in sustained and transient arousals were identified by computing a time-binned, paired $t$-test. The fMRI signal was averaged across 1-s bins, and the $p$ value was Bonferroni corrected for the number of time bins and ROIs tested. We then repeated the cross-correlation and bootstrap analyses in individual thalamic nuclei as described above, but we restricted the cross-correlation to only positive correlation values, since the positive correlation between thalamic regions had been established.

### Breathhold cerebrovascular reactivity measurements—Experiment 3

We used a breathhold task to measure fMRI signals caused by a vascular response to test whether the fMRI sequences observed during arousal state changes are due to hemodynamic latency differences[76,77,88]. When subjects hold their breath, the brain compensates for the reduction in oxygen by increasing blood flow to the brain, and when the breathhold is released, there is a momentary increase in fMRI signal due to the increased vasodilation. The dynamics during this vascular response were compared to the dynamics during arousal to test whether the results were confounded by inherent differences in vascular reactivity.

A projector displayed instructions for subjects to breathe freely for 27 s, followed by three paced breaths of 3 s in and 3 s out, then to hold their breath for 15 s. This cycle was repeated eight times per run. Four to eight runs were collected per subject. Breathhold releases were confirmed using simultaneously recorded respiratory signals. These scans underwent the same preprocessing steps as the sleep scans: motion correction, slice-time correction, and physiological noise removal. Breathholds with excessive motion (>0.3 mm) were excluded. Timeseries were extracted 20 s before and 20 s after the subject released their breathhold and began to breathe freely to capture both the beginning and end of the breathhold. ROI mean signals were averaged over this period, and the cross-correlation between regions was calculated. The hierarchical bootstrap analysis was repeated to generate 95% confidence intervals. These results were compared to the mean fMRI signal of ROIs during arousal state transitions. To correct the hemodynamic latencies based on the breathhold response, we subtracted the average breathhold lag from the behavioral arousal lag of each nucleus.

## Physiological analysis

To investigate how respiration and heart rate change during arousal, we calculated the mean amplitude of the respiratory and cardiac signals. The respiratory signal was bandpass filtered between 0.16–0.4 Hz, and the pulse signal was bandpass filtered between 0.7–1.8 Hz. Then, the instantaneous amplitude was calculated using the absolute value of the Hilbert transform and was averaged across all arousals.

## Reporting summary

Further information on research design is available in the Nature Research Reporting Summary linked to this article.

## Data availability

The source data for each figure are provided within this paper in the Source Data files, and the region of interest timeseries data locked to behavioral arousal are provided in a publicly accessible repository: https://github.com/bsetzer96/thalamic_arousal_sequence_ repository. Source data are provided with this paper.

## Code availability

This study primarily used freely accessible software: Freesurfer, FSL, and AFNI. MATLAB functions were used to call these standard tools for data analysis. The code used to generate the lag values and their confidence intervals for each thalamic nucleus is available in a publicly accessible repository: https://github.com/bsetzer96/thalamic_arousal_ sequence_repository.

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

## Acknowledgements
We are grateful to Sydney Bailes, Dr. Thomas Witzel, Azma Mareyam, and Dr. Donald Straney for their contributions to imaging and technical help. This study was funded by National Institutes of Health Grants R00-MH111748, R01-AG070135, R01-EB019437, U19-NS123717, P41-EB300006, S10-OD010759, the NARSAD Young Investigator Award from the Brain and Behavior Research Foundation, the Searle Scholars Program, the One Mind Rising Star Award, the Alfred P. Sloan Fellowship, the McKnight Scholar Award, the Pew Biomedical Scholar Award, and a Hariri Institute for Computing Fellowship to B.S.

## Author contributions
Conceptualization: B.S. and L.D.L. Data collection and methods: B.S., L.D.L., N.E.F., D.E.P.G., S.D.W., G.B., J.R.P. Data analysis: B.S. Writing—original draft: B.S. and L.D.L. Writing—review and editing: B.S., L.D.L., N.E.F., D.E.P.G., S.D.W., G.B., and J.R.P.

## Competing interests
The authors declare no competing interests.
