## [Peer Review File · Nature Communications]

A temporal sequence of thalamic activity unfolds at transitions in behavioral arousal stateREVIEWER COMMENTS

Reviewer #1 (Remarks to the Author):

This paper studied the brain-wide dynamics at behaviorally defined arousals. Concurrent fMRI and EEG signals were collected and analyzed at behavioral arousals defined as the button pressing after a period of inactivity in overnight experiments. It was found that the cortex and thalamus showed distinct dynamics at the behavioral arousals. The thalamus increased its activity seconds prior to the arousals, whereas the cortical regions were mainly characterized by strong deactivations ~10 seconds after the arousal. Importantly, close inspection of dynamics of thalamic nuclei revealed a stereotypic sequence of fMRI changes. The fMRI responses to breath holding were also examined to exclude the possibility that the sequential changes in the thalamus were due to the differences in the hemodynamic delays. Finally, the study also compared the fMRI responses at the transient and sustained arousals, and found that the thalamus only showed strong deactivations at the transient arousals. Overall, this is an interesting study that would provide some insight into brain network dynamics at the transitions of brain states. The paper is well written, and the results are clearly presented. Multiple experiments were conducted at different MRI scanners to increase scientific rigor. With that being said, I do have a few questions and comments as detailed below.

1. The sequential fMRI changes have been linked to transient arousal changes (Gu, et.al., Cerebral Cortex, 2021). In the paper, the global propagating waves in the cortex were studied and suggested to be linked to transient arousal modulations. The thalamus also showed sequential modulations across different nuclei at these cortical waves (Figure 6G in that paper). A simple comparison suggests some correspondence between the thalamus sequence associated with the cortical waves and that reported here, e.g., the CM and AV are sitting at the two ends of the sequences. Since the major finding of this paper is claimed to be the temporal sequence of thalamic activity in transitions in arousal states, the above-mentioned paper should be included at least for discussion.

2. The division of data into the thalamus and cortex of the most of analyses is somewhat arbitrary. I understand the authors want to emphasize the distinct dynamics of the two and thus a special role of the thalamus. However, it seems that it is a brain network including the thalamus and a few cortical regions that showed the early activation that was absent in the rest of the cortical regions. In the supplementary figure 2, there are quite a lot ROIs showing the activations according to the rough estimation based on the mean and standard error, they may not be statistically significant due to a strict standard.

The paper repeatedly emphasized the contrast between the thalamic activation and cortical deactivation to suggest/imply their distinct dynamics. But the fact is that the thalamus also showed the significant deactivation ~10 seconds after the behavioral arousals (Figure 2), just like the cortical regions.

As discussed above, some of the cortical regions also showed significant activations prior to the arousals along with the thalamus.

3. According to the supplementary figure 8, the thalamus sequence was only present at the transient behavioral arousal but not the sustained one. This is a strange result deserving more attention and discussion in the paper. The authors may also want to modify the paper, particularly the title and abstract, to make this clear. The current version reads like that this temporal sequence of thalamus appears at all the behavioral arousals, but it actually occurs only at certain type (transient) of behavioral arousals.

4. Another question is the potential confounding effect of the button pressing actions. This behavioral confounding is of less concern for the major findings, since the button press is not expected to induce widespread cortical changes as seen in the study and the analysis of thalamic activity was focused on the period prior to the arousal/button pressing. However, it might be significant for the result in Figure 6. Compared with the transient arousals, the sustained arousals were associated with more button pressing activations at the pace of breath. Is it possible that the less de-activated thalamus was partly due to the sustained activations due to the button pressing tasks? The authors may want to check the cortical regions responsible for sensorimotor functions to see whether a similar difference exist between the two types of arousals.

5. The definition of the latency of fMRI signal changes is essential for this study but quite tricky. It was identified as the time at which the fMRI signal reach 10% of its maximum. Would it depend on the peak fMRI amplitude? If the authors randomly split the subjects into two halves and repeat the analysis of Figure 4, would they get similar results?

Reviewer #2 (Remarks to the Author):

General comments

The manuscript by Setzer et al addresses an interesting and important question about the temporal sequence of thalamic and cortical responses to arousal. The study makes good use of ultra-fast fMRI, with good analyses and controls (eg looking at the impact of hemodynamic response variability on the temporal sequence within the thalamus). The manuscript itself is generally clearly written, although the Discussion in particular could be more concise. My main concern is with the data, and the joining together of different data sets consisting of few participants even in total. There could generally be more clarity about the data and the analyses (some specific examples are given below), and the use of the term 'arousal' is not always precise given the absence of EEG in the majority of the data. Behavioral vs electrophysiological arousal is addressed in the Discussion, but could be clearer through the manuscript as in several places the impression is given that all data sets are studying the sleep-wake

transition. In the absence of EEG to define sleep and wake unambiguously, this cannot be said. The lack of task performance is not necessarily an indication that the participant is asleep, and the terms 'unresponsiveness and active behavior' as used in places would be more precise. Sometimes the emphasis on early thalamic response does not entirely fit the data (eg cACC and PCC activate before thalamus, but the first line of the Discussion says 'behavioral arousal is preceded by a temporal sequence of activity across thalamic nuclei, followed by widespread cortical deactivation'). An interpretation whereby arousal is driven by cACC and PCC would also presumably fit the data.

Specific comments

Methods

1. Data sets are small (n=6, n=13, n=8) and heterogeneous scanning protocols and tasks are used across the three data sets. Despite some consistency in results, the lack of EEG in particular makes the interpretation of the data more speculative.
2. Experiment 1 acquisition covers 'most of the brain'. This needs to be made more precise (ie which parts of the brain does it not cover, is that consistent between participants etc).
3. Data consists of 'up to 90 minutes per run' – what is a run here, and did they do more than one run?
4. It would help the flow if 'Physiological recordings and analysis' was moved above 'fMRI Preprocessing', so that it is clear in the latter section that cardiac and respiratory signals were acquired.
5. fMRI data were oversampled to four times the original sampling rate. Was this with linear interpolation? If the test to see if there was a hemodynamic response prior to arousal looks at 1s bins, why do the oversampling to 62ms?
6. PCA was done to identify major activation patterns, but this is lacking in detail (num. components, how selected etc).
7. Analysis of whether it matters if subjects remained awake or fell asleep after arousal is presumably only in data set 1 with EEG (in expt 2 only 4 subjects have EEG)?

Results

1. Only expt 1 is able to say anything about transitioning between 'sleep and wakefulness' as the others don't have EEG.
2. In several places it is unclear which data are being referred (ie expt 1/2/3)
3. 'clearly from deeper stages of sleep (N2 and N3)' is this based on EEG (as it must be by definition), in which case it must only refer to a subset of the data where EEG was recorded?
4. 30 cortical regions from the Desikan-Killiany atlas – how were these regions selected when this atlas has considerably more than 30 regions?

5. As discussed above, cACC and PCC activate ~2s before the thalamus (-4.5/4.9s vs -2.8s), which is something at odds with the general conclusions and focus of the paper (in addition, does Fig 2a 'Cortex' include these regions?)
6. 'inter-nucleus correlation was higher during behavioral arousal than during wakefulness', it is unclear where the p value comes from (it is used to support the point that correlations are different, but unclear what test was used).
7. Fig 4: if the thalamus activates 2-3s before arousal as shown in the main analyses, and lag analysis in 4a shows that CM nucleus activates ~2s prior to the thalamus as a whole, how can the onset of CM be -8.5s in 4d?

Discussion

1. The sentence 'activity originates in an intralaminar nucleus and subsequently spreads to other thalamic nuclei' could do with clarification – presumably it is more likely that intralaminar nuclei receive stronger/faster input from brainstem, and in the absence of direct connections between nuclei the 'spread' could represent later brainstem input rather than a drive from intralaminar nuclei? This latter point also relates to the later sentence where activity is considered to be 'spreading to neighboring nuclei either directly', as the mechanism for this would presumably need to be via the thalamic reticular nucleus since there are not direct connection between thalamic nuclei?
2. The point about the cACC and PCC could be clarified ('as they are linked to effective thalamic modulation of behavioral arousal state') as with the data provided isn't it equally valid to have those cortical regions as the driver for arousal, since they are as early as thalamic regions?
3. 'the spike rate of neurons in the cortex is typically higher during wakefulness than during sleep', although reference 104 finds that upon awakening cortical neuronal firing is initially lower than baseline for several minutes?

Reviewer #3 (Remarks to the Author):

In this manuscript, the authors explore the role of different thalamic regions BOLD activity around the onset of awakening – assessed by a behavioural response that correlates with EEG signatures of arousal (increased alpha). They use fast 3T then 7T fMRI with a sub-second temporal resolution to quantify differences in temporal activations across the cortex and thalamus. They found that the thalamus begins its BOLD activation before behavioural onset and increases significantly following, whereas the cortex decreases mean BOLD levels following awakening. Then they explore specific thalamic and cortical regions demonstrating this is a general trend across the thalamic regions (albeit with different response times) and cortex (except for cingulate/pericalcarine regions). They cleverly validate the lagged thalamic activations after controlling hemodynamic response profiles, respiration, and heart rate. This reveals the centromedian thalamus precedes the other thalamic regions, which may be related to its observed role in arousal following stimulation. Finally, they find that increased thalamic activation is preserved for sustained versus transient waking and the temporal sequencing is a transient waking effect versus synchronised waking.

Overall, I find the paper is clearly written, and the experimental protocol and techniques are exceptional, however, insight into the mechanism remains unclear to me. Not that the results are not interesting, instead it is unclear the insight the authors believe this finding reveals.

Major comments:

1.

The significant finding of the paper is the lagged sequential thalamic activation or perhaps it is the coordinated activity given (suppFig.8); however, I am left wondering what the role of the lagged thalamic activation is for arousal, where it emerges, and the take-home message of the biological understanding for this lagged response is unclear to me.

1a. Regarding the importance, consistency between your finding of the rapid centromedian activation preceding awakening and stimulation of the centromedian awakening anaesthetised animals and the early (lag-corrected) anteroventral activation consistent with its role in alertness is very promising. However, how this relates to the sensory ventral posterolateral is less clear.

The final analysis comparing sustained/transient waking's this lagged activation is close to answering this question (Fig. S8). Would there be a way to compare if the lagged thalamic deactivation follows the same activation sequence or whether anything about the cortical activity differs dynamically between the two states (beyond mean activation)? Is the interpretation of Fig. S8 that the lagged hierarchy is not important at all?

1b. Regarding the second point, what may be the underlying cause of this sequential activation? For example, in the discussion, the concept of a propagating wave is suggested as a cause, but could this hypothesis be tested? One way to go about this would be to compare whether the same sequence is observed on single trials, as would be the case for a consistent travelling wave, or whether it is an averaging effect.

A comparison between your finding of global synchronisation and the recent findings of spatiotemporal BOLD patterns (Raut et al., 2021 10.1126/sciadv.abf2709) with arousal may aid this interpretation. From this result, we would expect lagged differences across the cortex depending on the spatial pattern.

2. A second major question I am left wondering is how I should interpret a deactivation of the cortex. It may reside outside the scope of this paper, as it has been shown already (ref 50 Zou et al., 2020).

However, it is unclear why the cortical signal is consistent between the motor and other cortical areas despite the continuation of a motor behaviour - button pressing.

The deactivation is very interesting as it is opposite of what one would expect following the increase of neural activity or vasodilation (via the increase in adrenergic neuromodulatory tone with waking (Aston-Jones and Cohen, 2005)). However, the activation observed in the Cingulate/Cuneus/Pericalcarine/Lingual vs the rest of the cortex supports that the results are not global vascular effects; nevertheless, these regions are all located adjacent to the medial cortical surface, which may play a role. Along these lines, how might I interpret the consistently deactivated cortex following transient arousals?

Perhaps some analysis into the thalamic and cortical dynamics following arousal may aid insight into beneficial properties of a 'deactivated' cortical brain. Furthermore, some example BOLD thalamic and cortical traces, at least in the supplementary figures, would be beneficial to understand the respective activation/deactivation.

3. How robust are the results between individuals? Both experiments have a large contribution from single individuals. The 7T results are based on 97 awakenings (1 person contributing 22), and 3T are based on 66 awakenings (1 person contributing 29). It is a testament to the authors that they have as many data points as they do, given the difficulty in getting patients to sleep in a scanner!

Minor

1. The analysis of changes relative to baseline are defined using the -10s window, are the results robust for different window sizes

2. Figures show standard error, is this 95% CI ($1.96 * SE$ or just 1 SE).

3. pg 7 paragraph 2: Please state onset time definition, ie 10% max, in text.

4. Discussion pg 11 paragraph 3, perhaps worthwhile citing/discussing the transition from burst firing of the thalamus to tonic with arousal as a possible source of cortical deactivation (Sherman 2001, TINS)

5. What were the average time between behavioural arousals? How many behavioural arousals did each individual have? More statistics about the waking's are needed.

RESPONSE TO REVIEWERS' COMMENTS

We really appreciate the thoughtful comments of all reviewers, who noted the innovation of the study and also raised several issues to address. We have revised the manuscript extensively (with the major new text additions highlighted in red, and several new figures) to fully resolve these issues, as discussed in detail below.

Reviewer #1 (Remarks to the Author):

This paper studied the brain-wide dynamics at behaviorally defined arousals. Concurrent fMRI and EEG signals were collected and analyzed at behavioral arousals defined as the button pressing after a period of inactivity in overnight experiments. It was found that the cortex and thalamus showed distinct dynamics at the behavioral arousals. The thalamus increased its activity seconds prior to the arousals, whereas the cortical regions were mainly characterized by strong deactivations ~10 seconds after the arousal. Importantly, close inspection of dynamics of thalamic nuclei revealed a stereotypic sequence of fMRI changes. The fMRI responses to breath holding were also examined to exclude the possibility that the sequential changes in the thalamus were due to the differences in the hemodynamic delays. Finally, the study also compared the fMRI responses at the transient and sustained arousals, and found that the thalamus only showed strong deactivations at the transient arousals.

Overall, this is an interesting study that would provide some insight into brain network dynamics at the transitions of brain states. The paper is well written, and the results are clearly presented. Multiple experiments were conducted at different MRI scanners to increase scientific rigor. With that being said, I do have a few questions and comments as detailed below.

1. The sequential fMRI changes have been linked to transient arousal changes (Gu, et.al., Cerebral Cortex, 2021). In the paper, the global propagating waves in the cortex were studied and suggested to be linked to transient arousal modulations. The thalamus also showed sequential modulations across different nuclei at these cortical waves (Figure 6G in that paper). A simple comparison suggests some correspondence between the thalamus sequence associated with the cortical waves and that reported here, e.g., the CM and AV are sitting at the two ends of the sequences. Since the major finding of this paper is claimed to be the temporal sequence of thalamic activity in transitions in arousal states, the above-mentioned paper should be included at least for discussion.

We are grateful to the reviewer for the helpful comments on our manuscript. We agree we should have cited this highly relevant paper, and we have now linked our results with Gu 2021¹ in the Discussion and also cited their manuscript in the Introduction.

2. The division of data into the thalamus and cortex of the most of analyses is somewhat arbitrary. I understand the authors want to emphasize the distinct dynamics of the two and thus a special role of the thalamus. However, it seems that it is a brain

network including the thalamus and a few cortical regions that showed the early activation that was absent in the rest of the cortical regions. In the supplementary figure 2, there are quite a lot ROIs showing the activations according to the rough estimation based on the mean and standard error, they may not be statistically significant due to a strict standard.

The paper repeatedly emphasized the contrast between the thalamic activation and cortical deactivation to suggest/imply their distinct dynamics. But the fact is that the thalamus also showed the significant deactivation ~10 seconds after the behavioral arousals (Figure 2), just like the cortical regions. As discussed above, some of the cortical regions also showed significant activations prior to the arousals along with the thalamus.

This is a great point, and we have now revised the language in several parts of the text to more clearly articulate that certain cortical regions are engaged in the pre-arousal sequence. First, we agree that not enough emphasis was placed on the fact that there is a subset of cortical regions that diverged from the dynamics across the whole cortex, and we have now emphasized the potential role of cACC and PCC in arousal state transitions in the Abstract and Discussion, including noting that we observe a sequence “a stereotyped sequence of activity across thalamic nuclei and cingulate cortex” in the Abstract. Additionally, we have added clarification in the “Thalamus activates before transitions in behavioral arousal state” section that the cingulate cortex activates, and also now describe that the thalamus also deactivates after arousal similarly to the cortex.

3. According to the supplementary figure 8, the thalamus sequence was only present at the transient behavioral arousal but not the sustained one. This is a strange result deserving more attention and discussion in the paper. The authors may also want to modify the paper, particularly the title and abstract, to make this clear. The current version reads like that this temporal sequence of thalamus appears at all the behavioral arousals, but it actually occurs only at certain type (transient) of behavioral arousals.

We apologize that we neglected to provide specific statistical information in this figure. We should have clarified that the sequence is statistically significant in the sustained case as well, with CM being significantly early for both types of arousals. The way that the confidence intervals were presented in this figure made this difficult to assess visually, because we had used an alternate method to compute lags intending for the supplementary figure to display an additional metric, which we were not sufficiently clear about (combining all pairs of lags within nuclei into a single metric, which is not readily visually interpretable). We had intended this supplementary figure to simply represent that pairwise sequences are similar overall, but it did not clearly display our result that an overall sequence is still preserved. We have now updated the confidence intervals to display the timing relative to the whole thalamus (i.e. using the same method as the main text), and have added the statistics and a new figure for this analysis (Fig. 7, S12). It can now be more clearly seen that CM is indeed significantly early in sustained arousals. However, we agree that the sequence is more compressed during sustained arousals. We have included additional description of this in the

“Thalamic signals differ in sustained and transient behavioral arousals” section of the Results.

4. Another question is the potential confounding effect of the button pressing actions. This behavioral confounding is of less concern for the major findings, since the button press is not expected to induce widespread cortical changes as seen in the study and the analysis of thalamic activity was focused on the period prior to the arousal/button pressing. However, it might be significant for the result in Figure 6. Compared with the transient arousals, the sustained arousals were associated with more button pressing activations at the pace of breath. Is it possible that the less de-activated thalamus was partly due to the sustained activations due to the button pressing tasks? The authors may want to check the cortical regions responsible for sensorimotor functions to see whether a similar difference exist between the two types of arousals.

We have now added a supplementary figure (Fig. S11) which shows that precentral cortex, which includes the primary motor cortex, does not show significantly different activity between sustained and transient arousals. We attribute this finding to the fact that the small fingertip motion is not sufficient to drive large signal changes in the larger parcels we examine. Additionally, we see significant differences in BOLD signal across the thalamus (Fig. S10), not just thalamic nuclei associated with motor action.

5. The definition of the latency of fMRI signal changes is essential for this study but quite tricky. It was identified as the time at which the fMRI signal reach 10% of its maximum. Would it depend on the peak fMRI amplitude? If the authors randomly split the subjects into two halves and repeat the analysis of Figure 4, would they get similar results?

The 10% onset time of thalamic nuclei may be influenced by the BOLD signal amplitude since it is defined as the time when the signal reaches 10% of the maximum amplitude from baseline. However, our paper demonstrates that using two different definitions of latency (both the cross-correlation in Fig.4a and onset time in Fig. 4d) both result in the same qualitative finding, that CM and VPL are earlier than all other nuclei. The cross-correlation analysis takes the full time-course into consideration and does not use peak amplitude to set thresholds, confirming that amplitude-based definitions are not required for this result. This result is therefore resilient to the precise definition of latency. We have added discussion of this in the “An activity sequence across thalamic nuclei prior to behavioral arousal” section of the Results.

We have also recomputed the thalamic activity sequence using a hierarchical bootstrap to explicitly take into account cross-subject variance as you suggest (Fig. S6a). The bootstrap analysis of lag time resamples from subjects first and then arousals. Therefore, the confidence intervals generated demonstrate that the lag analysis is resilient to variance across subjects and does not depend strongly on a small subset of subjects. Additionally, we repeated the lag analysis without the two subjects with the highest number of behavioral arousals and found the same sequence of lags in

the remaining 11 subjects (63 behavioral arousals, Fig. S6b). These two new analyses demonstrate that the latency analysis is reliable across the group.

Reviewer #2 (Remarks to the Author):

General comments

The manuscript by Setzer et al addresses an interesting and important question about the temporal sequence of thalamic and cortical responses to arousal. The study makes good use of ultra-fast fMRI, with good analyses and controls (eg looking at the impact of hemodynamic response variability on the temporal sequence within the thalamus). The manuscript itself is generally clearly written, although the Discussion in particular could be more concise. My main concern is with the data, and the joining together of different data sets consisting of few participants even in total. There could generally be more clarity about the data and the analyses (some specific examples are given below), and the use of the term 'arousal' is not always precise given the absence of EEG in the majority of the data. Behavioral vs electrophysiological arousal is addressed in the Discussion, but could be clearer through the manuscript as in several places the impression is given that all data sets are studying the sleep-wake transition. In the absence of EEG to define sleep and wake unambiguously, this cannot be said. The lack of task performance is not necessarily an indication that the participant is asleep, and the terms 'unresponsiveness and active behavior' as used in places would be more precise. Sometimes the emphasis on early thalamic response does not entirely fit the data (eg cACC and PCC activate before thalamus, but the first line of the Discussion says 'behavioral arousal is preceded by a temporal sequence of activity across thalamic nuclei, followed by widespread cortical deactivation'). An interpretation whereby arousal is driven by cACC and PCC would also presumably fit the data.

Thank you for your thoughtful feedback. We have streamlined the Discussion to assist in its conciseness. The remaining concerns are addressed in more detail below.

Specific comments

Methods

1. Data sets are small (n=6, n=13, n=8) and heterogeneous scanning protocols and tasks are used across the three data sets. Despite some consistency in results, the lack of EEG in particular makes the interpretation of the data more speculative.

We have now added new analyses to address concerns about group sizes and variance across subjects (Fig. S6a, b). This study uses within-subject analyses and does not use any whole-brain voxel-wise analyses, nor analyses of individual variability, to focus on questions that can be appropriately studied with high validity in a small number of subjects. In addition, due to the high temporal sampling of our acquisition, we acquire large amounts of data from each subject. While we agree that the n=6 of Experiment 1 is small, we designed Experiment 2 to test whether these results replicated in an independent dataset. We replicated the arousal-locked thalamic and cortical effects, with strikingly similar results in these two studies despite their distinct

acquisition parameters, and the key results that support our main conclusions are drawn from the n=13 dataset of Experiment 2. Our replication of Experiment 1 in the overall thalamic activity shows that this dataset is sufficient to achieve robust replicable effects, and our 95% confidence intervals show our effects are clearly significant within this sample. This number of subjects is also consistent with most prior fMRI studies of arousal states^{2,3} due to the large effect sizes of neural activity changes across arousal states. In Experiment 3, we use a different task because this comprises our control experiment, and we show 95% confidence intervals which are quite small, as our 7T imaging enables us to achieve high precision measurements within individuals. Therefore, the result is robust and does not require a large number of subjects.

We agree that we were not sufficiently clear in distinguishing behavioral arousal from the sleep to wake transition. We have revised in several places to address this concern, and added a sentence to the Introduction explicitly defining that our use of the term 'arousal' refers to behavioral arousal, as well as adding Discussion about the difference between behavioral vs. electrophysiological arousal. Additionally, we now make sure to use the terms "unresponsiveness" and "active behavior" rather than "sleep" and "wakefulness" as suggested, except in cases where we have confirmed with EEG. Our overall title aims to emphasize this point, clarifying that we are primarily analyzing transitions in behavioral arousal state.

2. Experiment 1 acquisition covers 'most of the brain'. This needs to be made more precise (ie which parts of the brain does it not cover, is that consistent between participants etc).

We have now added additional information about how the acquisition volume was placed in the 'BOLD fMRI acquisition' section of the Methods, specifying that "The acquisition volume was positioned consistently across subjects in an oblique angle, tilted to avoid capturing signal from the eyes, to minimize eye motion artifact. This positioning captured most of the brain but omitted the tip of the temporal lobe and the lower half of the cerebellum."

3. Data consists of 'up to 90 minutes per run' – what is a run here, and did they do more than one run?

We have expanded on the structure of the functional runs in the 'BOLD fMRI acquisition' section of the Methods, clarifying "This functional scan contained up to 1500 volumes and lasted up to 90 minutes per run. The run typically lasted 90 minutes if no other factors intervened but was sometimes shorter and followed by a subsequent run if scanning needed to be paused and restarted, for example if the subject pressed the squeeze ball or if a sensor needed to be adjusted."

4. It would help the flow if 'Physiological recordings and analysis' was moved above 'fMRI Preprocessing', so that it is clear in the latter section that cardiac and respiratory signals were acquired.

Thank you for your suggestion. We have moved the details of physiological data collection into the 'BOLD fMRI acquisition' section of the Methods.

5. fMRI data were oversampled to four times the original sampling rate. Was this with linear interpolation? If the test to see if there was a hemodynamic response prior to arousal looks at 1s bins, why do the oversampling to 62ms?

We included that the interpolation was indeed linear in the Methods and provided an explanation to why we oversample the data. Briefly, this interpolation allows for more accurate time-locking to behavioral arousal events, and reduces the temporal smearing of the activity across event-locked averages.

6. PCA was done to identify major activation patterns, but this is lacking in detail (num. components, how selected etc).

We have now added that the number of components was 39, as we did not reduce the number of components relative to the number of inputs. This is now specified in the Methods.

7. Analysis of whether it matters if subjects remained awake or fell asleep after arousal is presumably only in data set 1 with EEG (in expt 2 only 4 subjects have EEG)?

We apologize for the lack of clarity; this result was intended to represent whether the behavioral state was transient or sustained. We have now edited the language to specify that this comparison examines whether subjects continued behaving or directly went back to unresponsiveness, writing "Behavioral dynamics after arousal can vary substantially: the subject may fully transition and maintain an active behavioral state or may fall directly back into unresponsiveness after only a brief arousal."

Results

1. Only expt 1 is able to say anything about transitioning between 'sleep and wakefulness' as the others don't have EEG.

We agree with this point and have revised throughout the paper to address this concern and clarify when we analyze sleep vs. when we analyze behavioral arousal changes only. We have now added clarification in the introduction that we are focused on identifying dynamics underlying behavioral arousal, writing "We aim to identify the neural dynamics that underlie the recovery of behavior after a period of unresponsiveness, and refer to this transition in behavioral state as 'arousal'". We have also changed the language throughout the Results to specify when we are studying transitions from unresponsiveness to active behavior, vs. when we are specifically looking at sleep using the EEG. In addition, we added a paragraph to the Discussion which acknowledges the limitations of not having EEG data during each arousal and suggests future studies to compare behavioral arousal and electrophysiological arousal.

2. In several places it is unclear which data are being referred (ie expt 1/2/3)

We have now added a statement in the beginning of each section on which Experiment the data came from.

3. *'clearly from deeper stages of sleep (N2 and N3)' is this based on EEG (as it must be by definition), in which case it must only refer to a subset of the data where EEG was recorded?*

Yes, this analysis was done only in Experiment 1 data, which includes EEG for each subject. We have added clarification in the manuscript.

4. *30 cortical regions from the Desikan-Killiany atlas – how were these regions selected when this atlas has considerably more than 30 regions?*

There are 36 cortical regions in the Desikan-Killiany atlas, but because our acquisition volume does not capture the tip of the temporal lobe, we do not have data from each region in the atlas for every subject. Regions that were not captured in all subjects were not analyzed, to avoid confounds from varying numbers of subjects. We have now added descriptions of this in the Methods.

5. *As discussed above, cACC and PCC activate ~2s before the thalamus (-4.5/4.9s vs -2.8s), which is something at odds with the general conclusions and focus of the paper (in addition, does Fig 2a 'Cortex' include these regions?)*

We agree that cingulate cortex is also clearly engaged in the pre-arousal activity, and we should have discussed it in more detail. We have now added this in several places. We now note in the Abstract, "We discovered a stereotyped sequence of activity across thalamic nuclei and cingulate cortex that preceded behavioral arousal". We changed the language of the first sentence of the Discussion to include the early activity of the cACC and PCC with the thalamus. We also included this observation in the discussion: that cACC and PCC are linked with arousal since their activation is quite early. The overall Cortex measure did indeed include these regions; however, their contributions were relatively small since the other 28 cortical regions are also included in the 'Cortex' ROI.

6. *'inter-nucleus correlation was higher during behavioral arousal than during wakefulness', it is unclear where the p value comes from (it is used to support the point that correlations are different, but unclear what test was used).*

Thank you for pointing out this oversight. We used a paired t-test, and we have now added this to the text.

7. *Fig 4: if the thalamus activates 2-3s before arousal as shown in the main analyses, and lag analysis in 4a shows that CM nucleus activates ~2s prior to the thalamus as a whole, how can the onset of CM be -8.5s in 4d?*

In the latency analysis, we used a definition of when the thalamus reached 20% of its maximum absolute amplitude from baseline, which we had previously mistakenly reported as 10%, we apologize for the typo. We used a 20% cutoff for the latency definition for all analyses of mean fMRI signals because of the noise present when analyzing raw signals, rather than the smooth HRFs that were extracted for each thalamic nucleus to estimate onset times. We have now clarified this in the text. Because the latency is defined by when the mean signal itself reaches 20%, and the onset time of thalamic nuclei is defined by when their estimated smooth HRF reaches 10%, the latency will occur later in time. However, we have now calculated the onset time of the global thalamic signal as we did for each individual nucleus, and found that the onset time of the thalamus was -5.87 s. We have now included this in the Results. Therefore, the -8.5 s onset-time of CM is plausible, and this difference reflects that distinct methods for analyzing onsets will yield different absolute timing measures.

This comment highlights an overarching challenge, which is that the absolute times reported will always depend on the specific definition used to calculate timing. For this reason, our paper uses multiple metrics to confirm that multiple analyses all show the same relative timing sequence, even if absolute times differ. Each of these metrics are different and their scales can vary. For robustness, (i.e. to ensure that our particular definition of timing was not important) we measured the thalamic activity sequence in two different ways: activity onset, and relative activity lag. This allowed us to demonstrate that the relative timing is not sensitive to the method used. However, these two methods use different definitions, so the exact time of activity onset varies between them (since one defines onset, and the other defines relative timing across nuclei). The onset-time estimation is defined by a smooth representation of the thalamic nuclei, calculated by fitting an HRF to each nucleus. The lag analysis was done on the raw signals. Therefore, it is expected that the onset-times and lag analysis will be on different scales. However, we still see the same sequence across thalamic nuclei with both the onset-time and lag analysis methods. We have added clarification in the Results that this allowed us to determine that the thalamic sequence is robust to a specific method, and we added a specific definition of how onset-time was calculated.

Discussion

1. The sentence 'activity originates in an intralaminar nucleus and subsequently spreads to other thalamic nuclei' could do with clarification – presumably it is more likely that intralaminar nuclei receive stronger/faster input from brainstem, and in the absence of direct connections between nuclei the 'spread' could represent later brainstem input rather than a drive from intralaminar nuclei? This latter point also relates to the later sentence where activity is considered to be 'spreading to neighboring nuclei either directly', as the mechanism for this would presumably need to be via the thalamic reticular nucleus since there are not direct connection between thalamic nuclei?

Thank you for this helpful feedback. We agree that distinct inputs from brainstem are a very likely contributor to this observed sequence. We added additional discussion on how the activity could spread from CM to other thalamic nuclei, or from other brain regions.

2. *The point about the cACC and PCC could be clarified ('as they are linked to effective thalamic modulation of behavioral arousal state') as with the data provided isn't it equally valid to have those cortical regions as the driver for arousal, since they are as early as thalamic regions?*

We agree, and have now added clarification that cACC and PCC could also be contributing to modulating arousal transitions in the Abstract and Discussion.

3. *'the spike rate of neurons in the cortex is typically higher during wakefulness than during sleep', although reference 104 finds that upon awakening cortical neuronal firing is initially lower than baseline for several minutes?*

While the spike rate is below the awake baseline in the minutes after awakening, it is slowly increasing, and higher than during sleep. Therefore, the spike rate increases at transitions from sleep to wakefulness, even though it does not immediately fully recover to the awake baseline level. We added clarifying language about this statement.

Reviewer #3 (Remarks to the Author):

In this manuscript, the authors explore the role of different thalamic regions BOLD activity around the onset of awakening – assessed by a behavioural response that correlates with EEG signatures of arousal (increased alpha). They use fast 3T then 7T fMRI with a sub-second temporal resolution to quantify differences in temporal activations across the cortex and thalamus. They found that the thalamus begins its BOLD activation before behavioural onset and increases significantly following, whereas the cortex decreases mean BOLD levels following awakening. Then they explore specific thalamic and cortical regions demonstrating this is a general trend across the thalamic regions (albeit with different response times) and cortex (except for cingulate/pericalcarine regions). They cleverly validate the lagged thalamic activations after controlling hemodynamic response profiles, respiration, and heart rate. This reveals the centromedian thalamus precedes the other thalamic regions, which may be related to its observed role in arousal following stimulation. Finally, they find that increased thalamic activation is preserved for sustained versus transient waking and the temporal sequencing is a transient waking effect versus synchronised waking.

Overall, I find the paper is clearly written, and the experimental protocol and techniques are exceptional, however, insight into the mechanism remains unclear to me. Not that the results are not interesting, instead it is unclear the insight the authors believe this finding reveals.

Major comments:

1.
The significant finding of the paper is the lagged sequential thalamic activation or

perhaps it is the coordinated activity given (suppFig.8); however, I am left wondering what the role of the lagged thalamic activation is for arousal, where it emerges, and the take-home message of the biological understanding for this lagged response is unclear to me.

Thank you for your helpful feedback. Overall, we have rewritten including new text in Discussion paragraphs 2 and 3, to more clearly discuss the potential neural circuit implications of the dynamics we observe, using this approach to obtain simultaneous, fast measures across distributed thalamic nuclei, and have expanded on the implications of these results.

1a. Regarding the importance, consistency between your finding of the rapid centromedian activation preceding awakening and stimulation of the centromedian awakening anesthetised animals and the early (lag-corrected) anteroventral activation consistent with its role in alertness is very promising. However, how this relates to the sensory ventral posterolateral is less clear.

The final analysis comparing sustained/transient waking's this lagged activation is close to answering this question (Fig. S8). Would there be a way to compare if the lagged thalamic deactivation follows the same activation sequence or whether anything about the cortical activity differs dynamically between the two states (beyond mean activation)? Is the interpretation of Fig. S8 that the lagged hierarchy is not important at all?

We agree that it is unexpected that we found early activation in the ventral posterolateral nucleus of the thalamus. Though there are not prior studies identifying this nucleus as being involved in arousal state transitions, our results indicate that it may have a role in them. In our view, while early activity in CM might have been predicted (and supports that our method is an accurate way to measure thalamic dynamics), the VPL result was not known. This may reflect that animal studies must decide in advance which nucleus to record from, so VPL has not been examined in prior sleep studies. We have now added additional discussion of this novel finding and potential implications for understanding mechanisms regulating arousal. We now clarify that this is one of the insights provided by this study, identifying unexpected dynamics across nuclei, including the VPL.

We have also updated the sustained versus transient sequence figure, originally Fig. S8 in the first submission. We apologize for the lack of clarity, which was raised by Reviewer #1 as well. The way that the confidence intervals were presented in this figure made this difficult to assess visually because we had used an additional method to compute the lags' confidence interval in the supplementary figure, which we had intended as additional supporting evidence but were not sufficiently clear about (this older figure combined all pairs of lags between nuclei into a single metric, which is not readily visually interpretable). We now have added a main text figure which reports the overall lags for both arousal types (Fig. 7) and report the statistics showing that CM is consistently early both in sustained arousals and transient arousals. Because CM is still significantly early in sustained arousals, it is likely important for all arousal state transitions, regardless of their context. Since there is a larger spread across thalamic

nuclei during transient arousals, this suggests that the lag may be contributing to unresponsiveness. This lack of coordination across thalamic nuclei could inhibit the integration of information across the brain. This has been added to the Discussion.

As suggested, we have now also analyzed the thalamic deactivation sequence by identifying the moment that each nucleus begins to decrease. While there was a trend toward a sequence across the mean deactivation of each thalamic nucleus (Figure inset at right), beginning with CM and VPL, the confidence intervals overlapped substantially, so it was not significant. This highlights that the activation before behavioral arousals drives the sequence of activity across thalamic nuclei.

To investigate if any specific cortical activity differs in sustained and transient arousals, we have added a supplementary figure of the cortical subregions (Fig. S11). We observed that only one region (parahippocampal cortex) has significant differences between cortical regions during sustained and transient arousals. While we agree it would be really interesting to determine if any features of cortical activity can predict subsequent arousal state, given that there is no mean difference in these regions, it would require extensive new research to design and investigate what other features might be predictive of arousal, so in this manuscript we focus on differences between the mean BOLD signal, and we discuss in the Discussion potential aspects of cortical activity that may need further study (e.g. changes in synchrony rather than overall signal magnitude).

1b. Regarding the second point, what may be the underlying cause of this sequential activation? For example, in the discussion, the concept of a propagating wave is suggested as a cause, but could this hypothesis be tested? One way to go about this would be to compare whether the same sequence is observed on single trials, as would be the case for a consistent travelling wave, or whether it is an averaging effect.

A comparison between your finding of global synchronisation and the recent findings of spatiotemporal BOLD patterns (Raut et al., 2021 10.1126/sciadv.abf2709) with arousal may aid this interpretation. From this result, we would expect lagged differences across the cortex depending on the spatial pattern.

We have now rewritten and expanded the paragraph in the Discussion which considers potential mechanisms underlying the thalamic activity sequence. In addition to the traveling wave theory, we propose that activity may spread across thalamic nuclei through the thalamic reticular nucleus or cortex, or that it could be caused by weaker/late brainstem input to the late nuclei.

We have now tested if a propagating wave traversed the thalamus by analyzing the lag values during single trials as requested. We did not find a statistically significant activation sequence when we used this single-trial method, which we attribute to the high level of noise at the single trial level when analyzing tiny individual thalamic nuclei, as the individual trial traces had visible noise contributions and thus temporal

sequences could not be readily identified, which is typical for fMRI data (which generally requires averaging trials prior to analysis). Though we did not identify a traveling wave, our overall results are consistent with the order of activation seen in Gu 2021 and Raut 2021^{1,4}, and those studies also averaged over events and cycles. We have now included that a propagating wave could be a possible mechanism underlying thalamic lags in the discussion.

Additionally, we computed a lag analysis of the deactivation across the cortex (Fig. S8 and figure below, left panel). We found that there is a sequence of deactivation across cortical regions, beginning with pars cortical regions and ending with cingulate cortical regions (Fig. S8). This result is similar to Raut et al⁴ (right panel), but with noticeable differences. Specifically, they found that regions in the parietal cortex activate late, while our analysis showed parietal cortical lags to be close to zero. We have added a paragraph in the result section explaining this as well.

b Spatial spread of deactivation sequence

Raut et al. 2021. Global waves synchronize the brain's functional systems with

2. A second major question I am left wondering is how I should interpret a deactivation of the cortex. It may reside outside the scope of this paper, as it has been shown already (ref 50 Zou et al., 2020). However, it is unclear why the cortical signal is

consistent between the motor and other cortical areas despite the continuation of a motor behavior - button pressing.

The deactivation is very interesting as it is opposite of what one would expect following the increase of neural activity or vasodilation (via the increase in adrenergic neuromodulatory tone with waking (Aston-Jones and Cohen, 2005)). However, the activation observed in the Cingulate/Cuneus/Pericalcarine/Lingual vs the rest of the cortex supports that the results are not global vascular effects; nevertheless, these regions are all located adjacent to the medial cortical surface, which may play a role. Along these lines, how might I interpret the consistently deactivated cortex following transient arousals?

Perhaps some analysis into the thalamic and cortical dynamics following arousal may aid insight into beneficial properties of a 'deactivated' cortical brain. Furthermore, some example BOLD thalamic and cortical traces, at least in the supplementary figures, would be beneficial to understand the respective activation/deactivation.

Though we cannot pinpoint the exact cause of the cortical deactivation in both transient and sustained arousals, even in motor cortex, we can speculate on what may cause this. We added a supplementary figure of all cortical regions in sustained and transient arousals (Fig. S11), and there was no significant difference in precentral cortex. The button press task only involved one finger, and the precentral cortical region is very large, including the primary motor region for the whole body. Therefore, since the button-press would only affect a small region of the motor cortex, it may not be evident when analyzing the mean signal in the precentral gyrus. Thus, we did not observe a difference in BOLD signal in sustained and transient arousals, even though motor activity of the finger differs between the two cases. Secondly, the large effect of the behavioral transition event may overwhelm more subtle and specific motor activity. Thus, the deactivation of the cortex may reflect the moment of transition rather than the arousal state thereafter.

Though the caudal anterior cingulate and posterior cingulate activate before behavioral arousal, their BOLD signal decreases after behavioral arousal following the rest of the cortex. A decrease in thalamic BOLD signal also occurs at a similar point in time, following its early rise. Therefore, the decrease in BOLD signal following behavioral arousal could be a global vasculature effect due to systemic vasoconstriction at arousals via the sympathetic nervous system, or increased noradrenergic tone at awakening contributing to vasoconstriction. This has been clarified in the Results section of the text in 'Thalamus activates before behavioral arousal' and in the beginning of the Discussion.

We implemented additional analyses of the cortical deactivation following arousal. We analyzed the lags between cortical regions and found that a sequence of deactivation occurs across cortical regions, beginning with pars regions, and ending with cingulate regions (Fig. S8). We have included several examples of cortical and thalamic traces in the supplementary figures (Fig. S2, Fig. S4, Fig. S5, Fig. S8, Fig. S10, Fig. S11) to aid in the understanding of these dynamics.

3. How robust are the results between individuals? Both experiments have a large

contribution from single individuals. The 7T results are based on 97 awakenings (1 person contributing 22), and 3T are based on 66 awakenings (1 person contributing 29). It is a testament to the authors that they have as many data points as they do, given the difficulty in getting patients to sleep in a scanner!

We appreciate the reviewer's comment! The early rise in the thalamus followed by a decrease in the cortex is extremely robust, as it was replicated between Experiments 1 and 2. To test for the robustness of the thalamic lag sequence to subject variability, we computed a hierarchical bootstrap analysis, which resampled subjects and then their arousals (Fig. S6a). Since the 95% confidence intervals do not overlap for several nuclei, the activity sequence is robust across individuals. Additionally, we have now recomputed the lag analysis while excluding the two subjects with the most arousals in Experiment 2. We found that the activity sequence was largely unchanged with their removal (Fig. S6b). Therefore, our results are robust across individuals.

Minor

1. The analysis of changes relative to baseline are defined using the -10s window, are the results robust for different window sizes

We have recomputed the latency times for thalamus and cortex in Experiment 1 and 2 with varying window size and placements and found little change (Fig. S3).

*2. Figures show standard error, is this 95% CI (1.96*SE or just 1 SE).*

Figures that show standard error are just one standard error. Some figures show 95% confidence intervals, because the bootstrap analysis enables calculating the CI directly, and these are labeled. We have added an explicit definition of standard error in the Methods.

3. pg 7 paragraph 2: Please state onset time definition, ie 10% max, in text.

We have added the onset time definition to the text.

4. Discussion pg 11 paragraph 3, perhaps worthwhile citing/discussing the transition from burst firing of the thalamus to tonic with arousal as a possible source of cortical deactivation (Sherman 2001, TINS)

This is a good point, and we have added the thalamic transition and this reference in the Discussion.

5. What were the average time between behavioural arousals? How many behavioural arousals did each individual have? More statistics about the waking's are needed.

We have now added this information for both Experiment 1 and 2 in the supplementary information (Table S1, S2 respectively).

References

1. Gu, Y. *et al.* Brain Activity Fluctuations Propagate as Waves Traversing the Cortical Hierarchy. *Cereb. Cortex N. Y. N 1991* **31**, 3986–4005 (2021).
2. Larson-Prior, L. J. *et al.* Cortical network functional connectivity in the descent to sleep. *Proc. Natl. Acad. Sci.* **106**, 4489–4494 (2009).
3. Vahdat, S., Fogel, S., Benali, H. & Doyon, J. Network-wide reorganization of procedural memory during NREM sleep revealed by fMRI. *eLife* **6**, e24987 (2017).
4. Raut, R. V. *et al.* Global waves synchronize the brain's functional systems with fluctuating arousal. *Sci. Adv.* **7**, eabf2709 (2021).

REVIEWERS' COMMENTS

Reviewer #1 (Remarks to the Author):

The authors addressed all my comments in the previous run. I have no more comments and recommend the publication of this paper.

Reviewer #2 (Remarks to the Author):

The authors have provided a comprehensive response to the issues I raised, and have modified the manuscript accordingly. This has helped to clarify the work. I have no further issues to raise.

Reviewer #3 (Remarks to the Author):

I appreciate the authors' thoughtful revisions; the authors' responses and extended analysis have addressed most of my concerns, but two clarifications remain.

1. Being unfamiliar with the hierarchical bootstrap with replacement method (and perhaps other readers may be as well), a visual demonstration of the bootstrap distributions or a statement explaining how the mean lag can be near the upper (for Vpl/pul) or lower (VA/VLA) estimate of the Confidence Interval. I assume this means a strong non-Gaussianity to the lags (supporting the use of the bootstrapping vs Gaussian based statistics). Please discuss with clarification.
2. Consider uploading code utilised throughout the analysis to an online repository (GitHub etc.).

RESPONSE TO REVIEWERS 2

Reviewer #1 (Remarks to the Author):

The authors addressed all my comments in the previous run. I have no more comments and recommend the publication of this paper.

Reviewer #2 (Remarks to the Author):

The authors have provided a comprehensive response to the issues I raised, and have modified the manuscript accordingly. This has helped to clarify the work. I have no further issues to raise.

Thank you for your helpful feedback on our manuscript!

Reviewer #3 (Remarks to the Author):

I appreciate the authors' thoughtful revisions; the authors' responses and extended analysis have addressed most of my concerns, but two clarifications remain.

1. Being unfamiliar with the hierarchical bootstrap with replacement method (and perhaps other readers may be as well), a visual demonstration of the bootstrap distributions or a statement explaining how the mean lag can be near the upper (for Vpl/pul) or lower (VA/VLA) estimate of the Confidence Interval. I assume this means a strong non-Gaussianity to the lags (supporting the use of the bootstrapping vs Gaussian based statistics). Please discuss with clarification.

Thank you for your feedback. To clarify why the lag of the average timeseries does not always fall in the middle of the confidence interval (CI), we have included a supplementary figure which includes a histogram of lag values calculated from resampling using the bootstrap method (Fig. S6). It can now be seen that the distributions of the bootstrap resampled lag values are not always gaussian, and thus the lag value can fall closer to one end of the CI in some instances. To explain how the hierarchical bootstrap resampling was implemented, we have included a schematic which explains how the resampling is done (Fig. S7).

2. Consider uploading code utilised throughout the analysis to an online repository (GitHub etc.).

We will upload the code for the bootstrap resampling onto a publicly available repository in Github.